# Mixed Samples as Probes for Unsupervised Model Selection in Domain Adaptation

**Dapeng Hu**[1*] **Jian Liang**[3†] **Jun Hao Liew**[2] **Chuhui Xue**[2] **Song Bai**[2] **Xinchao Wang**[1†]

[1]National University of Singapore    [2]ByteDance Inc.

[3]CRIPAC & MAIS, Institute of Automation, Chinese Academy of Sciences

## Abstract

Unsupervised domain adaptation (UDA) has been widely applied in improving model generalization on unlabeled target data. However, accurately selecting the best UDA model for the target domain is challenging due to the absence of labeled target data and domain distribution shifts. Traditional model selection approaches involve training extra models with source data to estimate the target validation risk. Recent studies propose practical methods that are based on measuring various properties of model predictions on target data. Although effective for some UDA models, these methods often lack stability and may lead to poor selections for other UDA models. In this paper, we present MixVal, an innovative model selection method that operates solely with unlabeled target data during inference. MixVal leverages mixed target samples with pseudo labels to directly probe the learned target structure by each UDA model. Specifically, MixVal employs two distinct types of probes: the intra-cluster mixed samples for evaluating neighborhood density and the inter-cluster mixed samples for investigating the classification boundary. With this comprehensive probing strategy, MixVal elegantly combines the strengths of two state-of-the-art model selection methods, Entropy and SND. We extensively evaluate MixVal on 11 UDA methods across 4 adaptation settings, including classification and segmentation tasks. Experimental results consistently demonstrate that MixVal achieves state-of-the-art performance and maintains exceptional stability in model selection. Code is available at https://github.com/LHXXHB/MixVal.

## 1    Introduction

Despite the remarkable achievements of supervised learning in visual recognition tasks [1–4], deep neural networks face challenges when it comes to generalizing to out-of-distribution data [5]. As for the obstacle of out-of-domain generalization, deep domain adaptation techniques [6] provide an effective solution by transferring knowledge from a label-rich source domain to a related, yet label-scarce, target domain. Unsupervised domain adaptation (UDA) has drawn significant interest in recent years within the field of computer vision, including image classification [7–9], semantic segmentation [10–12], and object detection [13, 14], thanks to its practical setup with completely unlabeled target data. As the UDA landscape continues to evolve, various UDA settings have been explored to consider real-world scenarios, such as variations in category overlaps between domains and concerns related to source data privacy. These settings include closed-set UDA [7], partial-set UDA [15, 16], open-set UDA [17], open-partial-set UDA [18], and source-free UDA [19–21]. To effectively address these UDA problems, researchers have developed various novel solutions, including cross-domain alignment techniques [7, 9, 22, 11, 12], target-domain regularization methods [23–25], and self-training strategies in the target domain [26, 27, 20].

---

[*]Contact: lhxxhb15@gmail.com. Partial work was done during an internship at ByteDance with Song Bai.
[†]Corresponding authors: Jian Liang (liangjian92@gmail.com), and Xinchao Wang (xinchao@nus.edu.sg).

37th Conference on Neural Information Processing Systems (NeurIPS 2023).

Nevertheless, it is worth emphasizing that hyperparameters, including those associated with both the method-specific loss functions and the deep training pipeline, play a critical role in ensuring superior performance of UDA models during deployment. Notably, even the model trained by a state-of-the-art UDA approach can underperform the baseline source-trained model without adaptation if the hyperparameters are not properly set [28]. Surprisingly, the selection of hyperparameters has been largely overlooked in previous UDA studies, as highlighted by You *et al.* [18] and Saito *et al.* [29].

To further fill this research gap, this paper tackles the challenging problem of unsupervised model selection * in domain adaptation. Figure 1(a) illustrates the UDA validation pipeline. During the training stage, we employ different values of a hyperparameter $\beta$, denoted as $\beta_1, \beta_2, \beta_3$, to train corresponding UDA models denoted as $M_1, M_2, M_3$. Subsequently, in the validation stage, our objective is to identify the model with the minimal risk on the target domain and determine its associated $\beta$ value as the optimal choice for the UDA method. Dealing with model selection in UDA presents two primary challenges. On one hand, the absence of labeled target data makes conventional supervised target validation unfeasible. On the other hand, even if we had access to labeled source data, relying solely on the source risk [7] cannot ensure accurate estimation of the target risk due to domain distribution shifts between domains [29]. Therefore, advanced validation approaches are necessary to ensure accurate model selection in UDA.

Existing validation solutions in UDA can be classified into two types. The first type consists of source-based methods, which include the vanilla baseline of using source risk directly for target validation, denoted as SourceVal [7]. Additionally, Importance-Weighted Cross-Validation (IWCV)[30] and Deep Embedded Validation (DEV) [18] re-weight the source risk based on density similarity between domains. Reverse Validation (RV) [31] builds a symmetric UDA task to estimate target risk by reversing the source risk. However, source-based methods require training extra models involving source data and are vulnerable to severe distribution shifts between domains. On the other hand, the second type, target-only validation methods, is more straightforward and only utilizes unlabeled target samples. Entropy [30], based on the assumption of *low-density separation* [32], selects the model that produces target predictions with the lowest mean entropy. Soft Neighborhood Density (SND) [29], which relies on the assumption of *neighborhood consistency*, chooses the model with the highest consistency of predictions within the target neighborhood. While both Entropy and SND outperform source-based methods in certain UDA tasks, they have not fully explored the learned target structure for model selection, leading to instability across various UDA methods.

In this paper, we introduce MixVal, a novel target-only method for model selection in UDA. MixVal leverages mixed target samples as validation probes. To achieve this, MixVal employs *mixup* [33] with unlabeled target samples and their predictions, generating pseudo-labeled mixed samples. These mixed samples are further categorized into two types, depending on whether *mixup* is performed intra-cluster or inter-cluster. MixVal takes a novel strategy by evaluating inference-stage interpolation consistency for both types of mixed samples. Compared to Entropy and SND, MixVal innovatively employs mixed samples for probing the target structure, rather than directly measuring certain properties of target predictions. Moreover, MixVal elegantly combines the advantages of both Entropy and SND. On one hand, MixVal employs intra-cluster mixed samples to evaluate neighborhood density, leveraging the *neighborhood consistency* assumption utilized in SND. On the other hand, it utilizes inter-cluster mixed samples to evaluate the classification boundary, considering the *low-density separation* assumption used in Entropy. The novel probing allows MixVal to benefit from the strengths of both assumptions, making it a competitive UDA model selection method.

**Our main contributions** are highlighted as threefold:

- We study the significant yet under-explored model selection problem in UDA, providing comprehensive empirical evaluations of existing validation baselines.

- We introduce MixVal, a novel target-only validation method that directly probes the target structure using mixed samples. MixVal combines the strengths of both SND and Entropy through novel consistency-based probing with inter-cluster and intra-cluster mixed samples.

- We conduct extensive experiments to evaluate MixVal's effectiveness in model selection across various UDA settings, including closed-set UDA, partial-set UDA, open-partial-set UDA, and source-free UDA. The results highlight MixVal's exceptional stability and superior performance in UDA model selection compared to existing baselines.

---

*Throughout this paper, we use model selection, validation, and hyperparameter selection interchangeably.

Table 1: Comparison of assumptions considered in validation methods using target data.

| Validation Method | *neighborhood consistency* | no prior of class diversity | *low-density separation* |
|---|---|---|---|
| Entropy [42] | ✗ | ✓ | ✓ |
| InfoMax [28] | ✗ | ✗ | ✓ |
| SND [29] | ✓ | ✓ | ✗ |
| Corr-C [43] | ✗ | ✗ | ✓ |
| MixVal | ✓ | ✓ | ✓ |

## 2 Related Work

### 2.1 Unsupervised Domain Adaptation

Domain adaptation aims to leverage labeled source domains to facilitate transductive learning in a label-scarce target domain [34]. Unsupervised domain adaptation (UDA), which assumes that the target domain is entirely unlabeled, has gained significant attention due to its practical nature. Many traditional methods have been proposed to address UDA [35–38]. In recent years, deep learning-based UDA methods have witnessed significant progress, with domain adversarial learning emerging as a popular approach. Adversarial domain adaptation has been explored at multiple levels, including input-level [39], feature-level [7, 9, 15], prediction-level [11, 22], and entropy-level [12]. Additionally, there has been growing interest in adapting UDA models with a specific focus on the leverage of target data, leading to explorations in techniques such as target clustering [40, 27, 41] and target output regularization [24, 25]. Notably, Xu *et al.* [23] proposed a feature-level regularization approach that diverges from the popular mainstream methods. These UDA methods primarily address conventional scenarios that require source data for adaptation. More recently, a practical variant called source-free UDA has gained increasing attention [19, 20], where adaptation relies solely on the source-trained model. To comprehensively compare different validation approaches, we conduct hyperparameter selection for various representative UDA methods across diverse UDA settings.

### 2.2 Model Selection in Unsupervised Domain Adaptation

Model selection in UDA poses a significant challenge due to the absence of labeled target data and the presence of domain distribution shifts between domains. Some pioneering efforts have been made to address this issue. Ganin *et al.* [7] initially introduced the estimation of the unavailable target risk using source risk for target-domain validation (SourceVal). However, SourceVal is susceptible to severe domain shifts. Later, Ganin *et al.* [31] proposed Reverse Validation (RV), which considers a symmetric UDA problem from target to source and employs the reversed source risk for validation. Sugiyama *et al.* [30] introduced Importance-Weighted Cross-Validation (IWCV), which estimates the target risk by re-weighting the source risk based on input-level domain similarity. You *et al.* [18] further proposed Deep Embedded Validation (DEV), which considers feature-level similarity and controls variance in IWCV. While these source-based methods have proven effective, their practical applicability is limited by the need of extensive model training, access to source data, and the challenge of dealing with severe domain shifts. Consequently, recent efforts have shifted towards target-only validation methods, which rely solely on unlabeled target data. Morerio *et al.* [42] pioneered the use of mean entropy (Entropy) of target predictions for validation, considering the assumption of *low-density separation* [32]. Musgrave *et al.* [28] introduced the Information Maximization (InfoMax) score, which further considers *class diversity* in addition to Entropy. Tu *et al.* [43] introduced Corr-C, which prioritizes predictions with both large *class diversity* and high confidence. However, *class diversity* serves as a strict prior that is unsuitable for UDA scenarios with label shift [44]. Saito *et al.* [29] introduced Soft Neighborhood Density (SND), the state-of-the-art target-only validation method that prioritizes high *neighborhood consistency*. SND has demonstrated that target-only validation can outperform source-based methods, including IWCV and DEV. We also align with the research line of target-only validation, which offers simplicity and adaptability across various UDA settings. Table 1 showcases a comparison of how our method MixVal differs from other target-only validation methods in terms of the consideration of common assumptions.

## 2.3 Mixup

*Mixup*, a data augmentation technique originally proposed by Zhang *et al.* [33], is used during model training to improve generalization. *Mixup* has been extended to different levels, such as feature-level [45], patch-level [46], and token-level [47]. In addition to supervised learning, *mixup* has been successfully applied in semi-supervised learning [48, 49] and domain adaptation [50–52]. Unlike existing studies that consider *mixup* as a small perturbation to regularize model training, we employ *mixup* to generate in-between probing samples for model evaluation.

# 3 Methodology

## 3.1 Problem Setting

**Unsupervised domain adaptation (UDA).** We begin by introducing the UDA problem with a $K$-way image classification task, easily extendable to semantic segmentation tasks. In this setup, we are provided with a labeled source domain $\mathcal{D}_\mathrm{s} = \{(x_\mathrm{s}^i, y_\mathrm{s}^i)\}_{i=1}^{n_\mathrm{s}}$ consisting of $n_\mathrm{s}$ labeled images $x_\mathrm{s}$ with their corresponding labels $y_\mathrm{s}$, and an unlabeled target domain $\mathcal{D}_\mathrm{t} = \{x_\mathrm{t}^i\}_{i=1}^{n_\mathrm{t}}$ containing $n_\mathrm{t}$ unlabeled images $x_\mathrm{t}$. Here, $x_\mathrm{s}$ and $x_\mathrm{t}$ represent image vectors, and $y_\mathrm{s}$ are one-hot ground truth labels. The objective of UDA is to learn a discriminative model $M$ using $\mathcal{D}_\mathrm{s}$ and $\mathcal{D}_\mathrm{t}$, which can accurately predict the target labels $\{y_\mathrm{t}^i\}_{i=1}^{n_\mathrm{t}}$ for the corresponding target images $\{x_\mathrm{t}^i\}_{i=1}^{n_\mathrm{t}}$, under covariate shift [30] or label shift [44] between domains. Each image vector $x \in \mathcal{R}^d$ in the input space is associated with a one-hot pseudo label encoding, denoted as $\hat{y} \in \mathcal{R}^K$, which is predicted by model $M$. We denote $|\mathcal{C}_\mathrm{s}| = K$ as the number of categories in the source domain and $|\mathcal{C}_\mathrm{t}|$ as that in the target domain. Besides the vanilla UDA, which refers to closed-set UDA where both domains share the same label space ($\mathcal{C}_\mathrm{s} = \mathcal{C}_\mathrm{t}$), we also investigate other challenging UDA settings to evaluate the versatility of model selection approaches. These include partial-set UDA [15], where the source domain contains more categories than the target domain ($\mathcal{C}_\mathrm{s} \supset \mathcal{C}_\mathrm{t}$), and open-partial-set UDA [53], where both domains have overlapping but not identical labels ($\mathcal{C}_\mathrm{s} \cap \mathcal{C}_\mathrm{t} \neq \emptyset, \mathcal{C}_\mathrm{s} \not\supseteq \mathcal{C}_\mathrm{t}, \mathcal{C}_\mathrm{s} \not\subseteq \mathcal{C}_\mathrm{t}$).

Traditional UDA settings [34, 6, 15, 53] typically require simultaneous access to source and target data for effective target adaptation. The general optimization objective can be represented as follows:

$$\mathcal{L}_\mathrm{uda} = \mathcal{L}_\mathrm{src}(M; \mathcal{D}_\mathrm{s}) + \beta \mathcal{L}_\mathrm{adapt}(M; \mathcal{D}_\mathrm{s}, \mathcal{D}_\mathrm{t}, \eta).$$

Here, $\mathcal{L}_\mathrm{src}$ is the cross-entropy loss with labeled source data, and $\mathcal{L}_\mathrm{adapt}$ denotes the adaptation loss that adapts the training of model $M$ to unlabeled target data. The scalar coefficient $\beta$ is a common type of hyperparameter, and $\eta$ is a hyperparameter specific to the adaptation loss. Additionally, there are other training-related hyperparameters, such as learning rate, iterations, and network architecture.

Recently, source-free unsupervised domain adaptation (SFUDA) [19, 20] has gained significant attention in the research community. Unlike traditional UDA, SFUDA decouples the source supervised learning and unsupervised target adaptation into two sequential stages. The target adaptation stage in SFUDA is typically guided by the following objective function:

$$\mathcal{L}_\mathrm{sfuda} = \mathcal{L}_\mathrm{adapt}(M; M_\mathrm{s}, \mathcal{D}_\mathrm{t}, \eta).$$

Here, $M_\mathrm{s}$ represents the source pre-trained model. During the target adaptation stage, a target model $M$ is learned with access to the source model $M_\mathrm{s}$ and unlabeled target data $\{x_\mathrm{t}^i\}_{i=1}^{n_\mathrm{t}}$.

**Model selection.** In UDA, the goal of model selection is to pinpoint the hyperparameter configuration that offers the best performance on unlabeled target data. For example, let's consider the hyperparameter $\beta$, with a set of possible values $\{\beta_i\}_{i=1}^m$, where $m$ is the number of candidate values. We proceed to train $m$ distinct models $\{M_i\}_{i=1}^m$, each employing a different $\beta$ value. Our aim is to identify the model $M$ that delivers the highest performance on $\mathcal{D}_\mathrm{t}$ among all candidate UDA models $\{M_i\}_{i=1}^m$, thus determining the optimal hyperparameter value. Figure 1(a) visually illustrates this process, using 3 as the value for $m$. Model selection in UDA confronts two key challenges. Firstly, the absence of labeled target data renders traditional supervised validation methods, such as using a hold-out dataset [54], unfeasible. Secondly, the presence of severe domain distribution shifts between domains poses a risk when using source data for selecting the best model for the target data. Additionally, concerns about source data privacy can limit direct source data utilization.

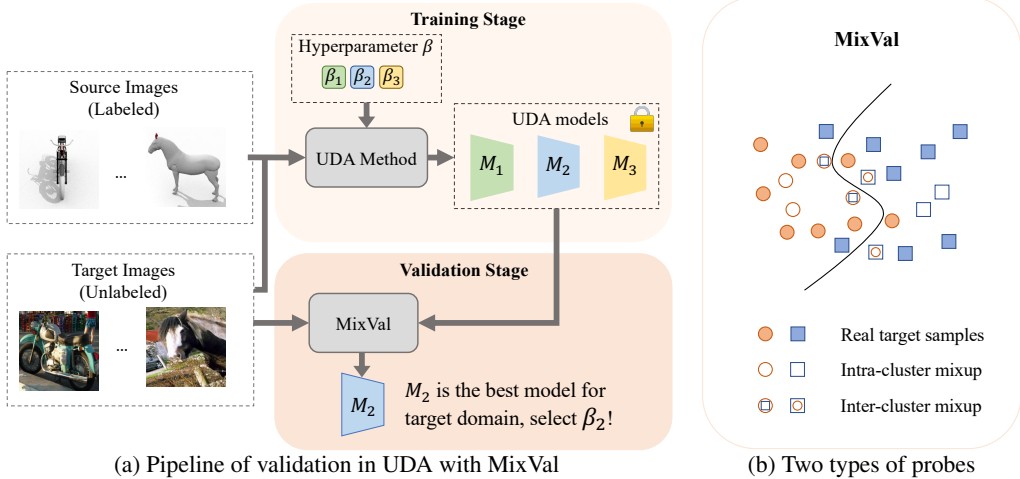

| (a) Pipeline of validation in UDA with MixVal | (b) Two types of probes |

Figure 1: Overview of the proposed method (MixVal) for model selection in UDA.

## 3.2 MixVal: Mixed Samples as Probes for Unsupervised Validation

**Motivation.** While existing target-only model selection methods [42, 29] have demonstrated competitive performance, they sometimes result in extremely poor selections when applied to different UDA methods and datasets, even within the standard closed-set UDA scenario. This phenomenon is substantiated by a comprehensive large-scale empirical study conducted by Musgrave *et al.* [28]. We highlight two common limitations in existing target-only validation methods that potentially hinder their ability to generalize across various UDA scenarios. First, these methods primarily rely on raw target predictions to calculate specific metrics for evaluating the learned target structure. However, it's important to note that these target data have already been utilized during the transductive model training. Second, while both the assumptions of *low-density separation* and *neighborhood consistency* are effective for model selection, existing validation methods often design their metrics based solely on one of these assumptions, as summarized in Table 1.

To address these limitations, we present MixVal, a target-only validation method that thoroughly evaluates the target structure through the use of mixed samples. We employ the *mixup* technique [33] to create novel in-between target samples. These mixed samples are then used for consistency evaluation to probe the learned target structure. With two distinct types of mixed samples, we ensure a comprehensive assessment of the target structure acquired by each candidate UDA model.

**Generation of mixed samples.** We first generate a labeled set of mixed samples using *mixup* with unlabeled target data and inference-stage UDA models. Specifically, we create a mixed sample $x_{\text{mix}}$ and its label $y_{\text{mix}}$ by performing a convex combination of a pair of target samples $x_t^i, x_t^j$ and their corresponding pseudo labels $\hat{y}_t^i, \hat{y}_t^j$ which are predicted by the UDA model. The process of mixed sample generation is formulated as follows:

$$x_{\text{mix}} = \lambda * x_t^i + (1 - \lambda) * x_t^j, \qquad y_{\text{mix}} = \lambda * \hat{y}_t^i + (1 - \lambda) * \hat{y}_t^j.$$

where $\lambda$ is a scalar used for interpolation, and $\hat{y}_t^i$ and $\hat{y}_t^j$ denote the one-hot pseudo label encodings. By performing *mixup* on random target samples for a single epoch in the inference stage, we generate a set of mixed samples $\{(x_{\text{mix}}^i, y_{\text{mix}}^i)\}_{i=1}^{n_t}$, where $n_t$ represents the total number.

**Interpolation consistency evaluation.** Next, we leverage each candidate UDA model to infer labels for all mixed samples, resulting in predicted labels $\{\hat{y}_{\text{mix}}^i\}_{i=1}^{n_t}$. Using both the mixed labels $\{y_{\text{mix}}^i\}_{i=1}^{n_t}$ and predicted labels, we assess the accuracy of mixed samples for each candidate model. This assessment yields the Interpolation Consistency Evaluation (ICE) score, defined as follows:

$$\text{ICE} = \text{Accuracy}(\{y_{\text{mix}}^i\}_{i=1}^{n_t}, \{\hat{y}_{\text{mix}}^i\}_{i=1}^{n_t}).$$

For all UDA models $\{M_i\}_{i=1}^{m}$, we apply the same *mixup* operation to generate identical mixed samples, but the mixed labels differ because model predictions for target samples vary among different models. We subsequently calculate the ICE score for each candidate model $M_i$, yielding a set of ICE scores $\{\text{ICE}_i\}_{i=1}^{m}$. The candidate model with a higher ICE score is deemed superior.

**MixVal via two types of probes.** Through a meticulous analysis of the *mixup* operation applied to unlabeled target samples and their pseudo labels, we have identified two distinct categories of mixed samples, as illustrated in Figure 1(b). The first type involves mixing target samples with the same pseudo label, referred to as intra-cluster *mixup*. When we evaluate the ICE score after performing intra-cluster *mixup*, we effectively measure the neighborhood density within each cluster. A higher ICE score indicates a higher level of *neighborhood consistency* [29]. Conversely, the second type involves mixing target samples with different pseudo labels, termed inter-cluster *mixup*. By evaluating the ICE score in this context, we can assess the classification margin between clusters. A higher ICE score in inter-cluster *mixup* signifies a larger margin of the classification boundary, indicating *low-density separation* [42]. For comprehensive probing, MixVal uses both types of mixed samples as probes. It separately ranks all candidate models in ascending order based on each type of ICE score. Then, MixVal takes the average rank from both rankings to select the candidate model with the highest average rank. Kindly refer to Appendix A for the pseudocode of MixVal.

**Connection to interpolation consistency training.** We provide a comprehensive comparison between our MixVal approach and interpolation consistency training (ICT), a commonly used technique in semi-supervised learning [48]. While our ICE score draws inspiration from ICT, there are three significant distinctions between MixVal and ICT. ICT primarily employs inter-cluster *mixup* as a minor training perturbation to regularize model learning with unlabeled data. In contrast, MixVal explicitly applies *mixup* to create in-between target samples, encompassing both inter-cluster and intra-cluster scenarios, during the inference stage without requiring model re-training.

# 4 Experiments

## 4.1 Setup

**Datasets.** For image classification tasks, we consider 4 popular UDA benchmarks of varied scales. *Office-31* [55] is a classic domain adaptation benchmark consisting of 31 object categories across 3 domains: Amazon (A), DSLR (D), and Webcam (W). *Office-Home* [56] is a challenging benchmark with 65 different object categories in 4 domains: Art (Ar), Clipart (Cl), Product (Pr), and Real-World (Re). *VisDA* [57] is a large-scale benchmark for the synthetic-to-real object recognition task, featuring 12 categories. It consists of a training (T) domain with 152k synthetic images and a validation (V) domain with 55k realistic images. *DomainNet* [58] is a recent large-scale benchmark comprising approximately 600k images across 345 categories in 6 distinct domains. For evaluation, we focus on a subset of 126 classes with 7 tasks [59] from 4 domains: Real (R), Clipart (C), Painting (P), and Sketch (S). For semantic segmentation tasks, we use the synthetic *GTAV* [60] dataset as the source domain and the real-world *Cityscapes* [61] dataset as the target domain.

**UDA methods.** We use the validation baselines discussed in Section 2.2 to conduct model selection for various UDA methods, with a specific emphasis on the practical target-only baselines. For closed-set UDA, we consider ATDOC [27], BNM [24], CDAN [9], MCC [25], MDD [22], SAFN [23], DMRL [50], AdaptSeg [11], and AdvEnt [12]. For partial-set UDA, we consider PADA [15] and SAFN [23]. For source-free UDA, we consider SHOT [20]. For open-partial-set UDA, we consider DANCE [41]. For ATDOC, BNM, CDAN, PADA, SAFN, SHOT, DMRL, and DANCE, we select the loss coefficient among 7 varied candidate values. For MDD, we validate the margin factor, while for MCC, we validate the temperature. For AdaptSeg and AdvEnt, we validate both the loss coefficient and training iteration. For MCC and MDD, we also include a two-hyperparameter validation task, where the bottleneck dimension is considered as an additional hyperparameter, with 4 possible values. Kindly refer to Appendix B for details of hyperparameter settings.

**Implementation details.** We utilize the Transfer Learning Library[†] to train UDA models on a single RTX TITAN 16GB GPU. The batch size is set to 32, and the total number of iterations is set to 5,000. We save the final model as a checkpoint. For *VisDA* and *GTAV2Cityscapes*, we use ResNet-101 [3], for *DomainNet*, we use ResNet-34 [3], and for the other benchmarks, we use ResNet-50 [3]. Regarding SND [29], we employ the official implementation. For source-based methods, we split 80% of the source data as the training set and the remaining 20% as the validation set. In our MixVal approach, we use a fixed value of $\lambda$ to ensure a fair comparison among candidate models. Specifically, we set $\lambda = 0.55$ for all experiments, which allows us to probe with in-between samples.

---

[†] https://github.com/thuml/Transfer-Learning-Library

Table 2: Validation accuracy (%) of closed-set UDA on *Office-Home* (*Home*). **bold**: Best value.

| Method | ATDOC [27] | | | | | BNM [24] | | | | | CDAN [9] | | | | |
|---|---|---|---|---|---|---|---|---|---|---|---|---|---|---|---|
| | →Ar | →Cl | →Pr | →Re | avg | →Ar | →Cl | →Pr | →Re | avg | →Ar | →Cl | →Pr | →Re | avg |
| SourceVal | 66.63 | 52.54 | 78.57 | 76.61 | 68.59 | 62.44 | 50.74 | 77.53 | 74.76 | 66.37 | 55.00 | 42.65 | 69.50 | 68.81 | 58.99 |
| IWCV [30] | 67.97 | 54.03 | 78.31 | 79.26 | 69.89 | 66.56 | 48.16 | 74.09 | 73.28 | 65.52 | 61.31 | 41.24 | 67.17 | 71.93 | 60.41 |
| DEV [18] | 67.39 | 54.23 | 77.78 | 79.39 | 69.70 | 65.76 | 56.39 | 73.92 | 77.59 | 68.41 | 67.23 | 57.04 | 68.76 | 76.91 | 67.49 |
| RV [31] | 68.68 | 56.13 | **78.93** | **79.64** | **70.85** | **68.25** | **56.75** | **78.08** | **78.67** | **70.44** | 67.66 | 56.74 | 76.01 | 77.68 | 69.52 |
| Entropy [42] | 63.67 | 55.83 | 76.54 | 78.36 | 68.60 | 66.28 | 54.49 | 74.15 | 77.64 | 68.14 | 67.66 | 57.56 | 76.37 | 77.45 | 69.76 |
| InfoMax [28] | 63.67 | 55.63 | 77.61 | 78.36 | 68.82 | 66.28 | 54.49 | 74.15 | 77.64 | 68.14 | 67.66 | 57.56 | 76.37 | 77.45 | 69.76 |
| SND [29] | 63.67 | 55.63 | 76.54 | 77.54 | 68.34 | 66.28 | 54.49 | 74.15 | 77.64 | 68.54 | **67.94** | 57.56 | **76.96** | 77.68 | **70.04** |
| Corr-C [43] | 63.51 | 50.39 | 73.89 | 73.88 | 65.42 | 58.10 | 45.37 | 68.97 | 70.59 | 60.76 | 53.84 | 41.21 | 64.96 | 67.65 | 56.91 |
| MixVal | 66.47 | **56.87** | 78.14 | 79.20 | 70.17 | 67.36 | 56.18 | 76.10 | 78.12 | 69.44 | 67.71 | **57.78** | 76.89 | **77.76** | 70.03 |
| Worst | 62.89 | 50.39 | 73.89 | 73.88 | 65.26 | 58.10 | 45.37 | 68.96 | 70.59 | 60.75 | 53.80 | 41.21 | 64.78 | 67.65 | 56.86 |
| Best | 68.97 | 58.35 | 80.27 | 80.58 | 72.04 | 68.93 | 57.51 | 78.43 | 79.57 | 71.11 | 68.19 | 57.90 | 77.44 | 78.19 | 70.43 |

| Method | MCC [25] | | | | | MDD [22] | | | | | SAFN [23] | | | | | *Home* AVG |
|---|---|---|---|---|---|---|---|---|---|---|---|---|---|---|---|---|
| | →Ar | →Cl | →Pr | →Re | avg | →Ar | →Cl | →Pr | →Re | avg | →Ar | →Cl | →Pr | →Re | avg | |
| SourceVal | 66.57 | 56.53 | 79.55 | 80.90 | 70.89 | 62.53 | 54.43 | 75.27 | 75.55 | 66.94 | 63.54 | 51.34 | 73.66 | 74.54 | 65.77 | 66.26 |
| IWCV [30] | 68.69 | 58.93 | 80.37 | 80.08 | 72.02 | 64.20 | 56.50 | 73.78 | 74.28 | 67.19 | 64.31 | 52.36 | 72.31 | 74.29 | 65.82 | 66.81 |
| DEV [18] | 68.81 | 58.07 | 78.54 | 80.10 | 71.38 | 64.42 | 56.94 | 76.85 | 75.94 | 68.54 | 63.15 | 50.47 | 71.20 | 74.54 | 64.84 | 68.39 |
| RV [31] | **70.40** | 58.80 | 80.63 | 80.39 | 72.56 | **66.57** | 55.75 | 76.60 | 76.90 | 68.96 | 64.31 | 50.13 | 73.77 | 74.93 | 65.78 | 69.68 |
| Entropy [42] | 69.29 | **59.33** | 80.63 | 80.96 | 72.55 | 66.54 | 57.63 | 77.27 | 77.45 | 69.72 | 59.85 | 46.41 | 72.51 | 73.18 | 62.99 | 68.63 |
| InfoMax [28] | 66.58 | 58.48 | 79.12 | 80.81 | 71.25 | 66.54 | 57.74 | 77.27 | 77.45 | **69.75** | 64.56 | 49.71 | 73.77 | 73.18 | 65.31 | 68.84 |
| SND [29] | 69.05 | 55.61 | 79.72 | 79.10 | 70.87 | 51.34 | 38.01 | **77.61** | 68.46 | 58.86 | 57.90 | 46.41 | 67.04 | 68.18 | 59.88 | 66.02 |
| Corr-C [43] | 69.05 | 55.61 | 79.72 | 79.10 | 70.87 | 47.79 | 31.69 | 63.40 | 60.63 | 50.88 | 62.66 | 46.41 | 68.83 | 68.18 | 61.52 | 61.06 |
| MixVal | 69.79 | 59.24 | 80.47 | 80.74 | 72.56 | 65.73 | **58.01** | 77.36 | 76.91 | 69.50 | **65.98** | **53.14** | **74.76** | **75.40** | **67.32** | **69.84** |
| Worst | 62.72 | 54.63 | 76.19 | 78.19 | 67.93 | 47.79 | 31.69 | 63.40 | 60.63 | 50.88 | 57.90 | 46.41 | 67.04 | 68.18 | 59.88 | 60.26 |
| Best | 70.68 | 59.95 | 80.93 | 81.02 | 73.14 | 66.75 | 58.36 | 77.61 | 77.45 | 70.04 | 66.59 | 53.14 | 74.90 | 75.57 | 67.55 | 70.72 |

Table 3: Validation accuracy (%) of closed-set UDA on *Office-31* (*Office*) and *VisDA*.

| Method | ATDOC [27] | | | | | BNM [24] | | | | | CDAN [9] | | | | |
|---|---|---|---|---|---|---|---|---|---|---|---|---|---|---|---|
| | →A | →D | →W | avg | T→V | →A | →D | →W | avg | T→V | →A | →D | →W | avg | T→V |
| SourceVal | 72.56 | 88.96 | **87.80** | 83.11 | 67.79 | 72.92 | **90.36** | **89.43** | 84.24 | 70.51 | 63.90 | 91.16 | **89.06** | 81.37 | 64.50 |
| IWCV [30] | 72.56 | 86.14 | 86.54 | 81.75 | 67.79 | 72.92 | 85.54 | **89.43** | 82.63 | **76.94** | 63.90 | 69.08 | 58.74 | 63.91 | 64.50 |
| DEV [18] | 72.56 | 86.14 | 86.54 | 81.75 | 70.34 | 72.92 | 85.54 | **89.43** | 82.63 | **76.94** | 63.90 | 91.16 | 88.30 | 81.12 | 64.50 |
| RV [31] | 74.93 | 89.96 | 87.23 | 84.04 | **77.37** | 70.71 | 88.55 | **89.43** | 82.90 | 74.58 | **73.27** | 91.16 | 88.30 | 84.24 | 76.02 |
| Entropy [42] | 73.29 | 86.14 | **87.80** | 82.41 | 62.85 | 72.67 | 85.54 | 83.14 | 80.45 | 58.36 | 71.62 | 91.16 | **89.06** | 83.95 | **80.46** |
| InfoMax [28] | 73.29 | 86.14 | **87.80** | 82.41 | 76.49 | 70.52 | 85.54 | 83.14 | 79.73 | 58.36 | 71.62 | 91.16 | 88.30 | 83.69 | **80.46** |
| SND [29] | 73.29 | 92.37 | **87.80** | 84.49 | **77.37** | 74.44 | 85.54 | 83.14 | 81.04 | 69.65 | 71.55 | 92.37 | 88.55 | 84.16 | **80.46** |
| Corr-C [43] | 71.05 | 90.96 | 84.40 | 82.14 | 67.79 | 67.16 | 84.34 | 78.99 | 76.83 | 70.51 | 67.67 | 67.67 | 59.62 | 61.86 | 64.50 |
| MixVal | 73.61 | 90.96 | 86.54 | 83.70 | **77.37** | 74.97 | 86.48 | 87.00 | 82.81 | 74.51 | 72.73 | **92.64** | **89.06** | **84.81** | **80.46** |
| Worst | 71.05 | 86.14 | 84.40 | 80.53 | 62.85 | 67.16 | 84.34 | 78.99 | 76.83 | 23.08 | 58.29 | 67.67 | 57.11 | 61.02 | 64.50 |
| Best | 75.31 | 92.37 | 87.80 | 85.16 | 77.37 | 75.52 | 90.36 | 89.43 | 85.10 | 76.94 | 73.38 | 92.77 | 89.06 | 85.07 | 80.46 |

| Method | MCC [25] | | | | | MDD [22] | | | | | SAFN [23] | | | | | *Office* AVG | *VisDA* AVG |
|---|---|---|---|---|---|---|---|---|---|---|---|---|---|---|---|---|---|
| | →A | →D | →W | avg | T→V | →A | →D | →W | avg | T→V | →A | →D | →W | avg | T→V | | |
| SourceVal | 73.11 | 90.96 | 91.07 | 85.05 | 80.46 | 75.72 | 91.06 | 86.23 | 84.34 | 72.25 | 69.20 | 83.73 | 87.17 | 80.03 | 70.71 | 83.02 | 71.04 |
| IWCV [30] | 73.11 | 91.16 | 88.55 | 84.27 | 81.48 | 75.49 | 91.16 | 89.18 | 85.28 | 72.25 | 69.32 | 86.55 | 80.38 | 78.75 | 66.33 | 79.43 | 71.55 |
| DEV [18] | 72.70 | 89.16 | 93.08 | 84.98 | 81.48 | 75.65 | 91.16 | 89.18 | 85.33 | 72.25 | 68.21 | 86.55 | 80.38 | 78.38 | 66.33 | 82.36 | 71.97 |
| RV [31] | 73.97 | 89.06 | 93.08 | 85.37 | **82.22** | 74.46 | **92.57** | 86.79 | 84.61 | 77.23 | 68.69 | 90.83 | 87.17 | 82.23 | 66.33 | 83.90 | 75.62 |
| Entropy [42] | 73.93 | 90.56 | **93.46** | 85.98 | **82.22** | 76.31 | **92.57** | 90.82 | 86.57 | **78.95** | 68.23 | **91.57** | 85.66 | 81.82 | 70.20 | 83.53 | 72.17 |
| InfoMax [28] | 73.63 | 89.16 | 88.55 | 83.88 | 81.48 | **76.50** | **92.57** | 90.82 | 86.63 | **78.95** | 68.23 | **91.57** | 87.42 | **82.41** | 70.20 | 83.13 | 74.32 |
| SND [29] | 73.93 | 91.97 | **93.46** | 86.45 | 69.35 | **76.50** | 92.17 | 90.82 | 86.50 | **78.95** | 68.23 | 89.96 | 85.66 | 81.28 | 58.15 | 83.99 | 72.32 |
| Corr-C [43] | 73.93 | 91.37 | **93.46** | 86.25 | 69.35 | 74.25 | 91.57 | 85.66 | 83.83 | 72.25 | 68.39 | 86.75 | 80.38 | 78.51 | 62.52 | 78.24 | 67.82 |
| MixVal | **74.09** | 91.77 | **93.46** | 86.36 | 81.48 | 75.97 | 91.74 | 90.42 | 86.49 | **78.95** | **69.61** | 89.96 | 86.83 | 82.13 | **74.41** | **84.39** | **77.86** |
| Worst | 70.56 | 86.75 | 87.17 | 81.49 | 69.35 | 73.06 | 87.35 | 85.66 | 82.02 | 72.25 | 67.27 | 83.73 | 80.38 | 77.13 | 58.15 | 76.50 | 58.36 |
| Best | 74.42 | 91.97 | 93.46 | 86.62 | 82.23 | 76.52 | 92.57 | 92.20 | 87.10 | 78.95 | 70.06 | 91.57 | 87.42 | 83.02 | 75.30 | 85.34 | 78.54 |

Table 4: Validation accuracy (%) of closed-set UDA on *DomainNet*.

| Method | CDAN [9] | | | | | BNM [24] | | | | | ATDOC [27] | | | | |
|---|---|---|---|---|---|---|---|---|---|---|---|---|---|---|---|
| | → C | → P | → R | → S | avg | → C | → P | → R | → S | avg | → C | → P | → R | → S | avg |
| Entropy [42] | **67.09** | **65.80** | **74.42** | **59.34** | **66.66** | 63.36 | 64.28 | 74.31 | 48.69 | 62.66 | 63.75 | 61.85 | **79.60** | 52.17 | 64.34 |
| InfoMax [28] | **67.09** | **65.80** | **74.42** | **59.34** | **66.66** | 67.05 | 64.28 | 74.31 | 55.67 | 65.33 | 63.75 | 61.85 | **79.60** | 52.17 | 64.34 |
| SND [29] | **67.09** | 64.68 | **74.42** | **59.34** | 66.38 | 56.56 | 54.50 | 74.31 | 42.37 | 56.93 | 63.75 | 61.85 | **79.60** | 47.00 | 63.05 |
| Corr-C [43] | 57.35 | 62.88 | **74.42** | 54.63 | 62.32 | 59.75 | 63.41 | 77.62 | 42.37 | 60.79 | 59.98 | 62.27 | 74.42 | 53.69 | 62.59 |
| MixVal | **67.09** | **65.80** | **74.42** | **59.34** | **66.66** | 67.84 | 66.40 | 78.68 | 58.49 | 67.85 | **68.94** | **68.44** | **79.60** | **61.73** | **69.68** |
| Worst | 57.35 | 60.76 | 73.44 | 51.41 | 60.74 | 55.79 | 54.50 | 74.31 | 42.37 | 56.74 | 59.98 | 61.85 | 74.42 | 47.00 | 60.81 |
| Best | 67.09 | 65.80 | 74.44 | 59.34 | 66.66 | 67.86 | 66.50 | 78.68 | 58.49 | 67.88 | 70.30 | 68.44 | 80.38 | 62.23 | 70.34 |

## 4.2 Results

We report MixVal's validation performance as averages from three random runs. In the tables, 'Best' signifies the optimal selection, and 'Worst' represents the worst one. For brevity, we report average results for tasks with the same target domain. Kindly refer to Appendix D for detailed results.

**Closed-set UDA.** We begin by comparing the performance of validation baselines within the standard closed-set UDA scenario. Specifically, we report the results for both source-based and target-only validation methods across several well-established UDA benchmarks. Table 2 provides the results on the medium-scale benchmark *Office-Home*, while Table 3 presents the results on the small-scale benchmark *Office-31* and the large-scale benchmark *VisDA*. Furthermore, Table 4 presents the results

Table 5: Validation accuracy (%) of partial-set UDA on *Office-Home*.

| Method | PADA [15] | | | | | SAFN [23] | | | | |
|---|---|---|---|---|---|---|---|---|---|---|
| | → Ar | → Cl | → Pr | → Re | avg | → Ar | → Cl | → Pr | → Re | avg |
| SourceVal | 57.21 | 41.90 | 64.48 | 71.89 | 58.87 | 66.82 | 54.71 | 74.41 | 76.48 | 68.11 |
| IWCV [30] | 59.65 | 50.51 | 66.84 | 72.96 | 62.49 | 69.36 | 53.91 | 71.78 | 76.38 | 67.86 |
| DEV [18] | 66.88 | 49.29 | 72.40 | 70.46 | 64.76 | 69.36 | 54.94 | 73.95 | 76.06 | 68.58 |
| RV [31] | 57.79 | 40.87 | 63.87 | 70.83 | 58.34 | 68.98 | 52.74 | 72.83 | 77.14 | 67.92 |
| Entropy [42] | 60.08 | 46.51 | 53.16 | 62.47 | 55.56 | **71.75** | 55.62 | 76.36 | 76.59 | 70.08 |
| InfoMax [28] | 60.08 | **51.40** | 60.20 | 66.67 | 59.59 | 63.67 | 51.74 | 69.64 | 73.62 | 64.67 |
| SND [29] | **67.80** | 50.71 | 59.46 | 67.13 | 61.27 | **71.75** | 51.74 | 76.36 | 78.36 | 69.55 |
| Corr-C [43] | 61.34 | 45.65 | 54.90 | 62.25 | 56.04 | 71.23 | 55.70 | 76.94 | **79.13** | 70.75 |
| MixVal | 67.68 | 51.01 | **72.94** | **78.64** | **67.57** | 71.70 | **57.91** | **77.08** | 78.94 | **71.41** |
| Worst | 56.29 | 39.76 | 50.49 | 59.31 | 51.46 | 62.48 | 49.91 | 68.50 | 73.62 | 63.63 |
| Best | 69.33 | 55.86 | 74.55 | 79.59 | 69.83 | 73.37 | 58.09 | 77.35 | 79.33 | 72.03 |

Table 6: HOS (%) of open-partial-set UDA and accuracy (%) of source-free UDA.

| Method | DANCE [41] | | | | *Home* | SHOT [20] | | | *Office* | *VisDA* |
|---|---|---|---|---|---|---|---|---|---|---|
| | →Ar | →Cl | →Pr | →Re | avg | →A | →D | →W | avg | T→V |
| Entropy [42] | 32.00 | 39.48 | 27.52 | 38.08 | 34.27 | 71.67 | 90.76 | 88.68 | 83.70 | 82.65 |
| InfoMax [28] | 32.00 | 39.48 | 27.52 | 38.01 | 34.25 | 71.67 | 90.76 | 88.68 | 83.70 | 82.65 |
| SND [29] | 15.05 | 4.33 | 23.75 | 16.79 | 14.98 | 71.67 | 90.76 | 88.68 | 83.70 | 82.65 |
| Corr-C [43] | 29.60 | 4.33 | 23.75 | 16.79 | 18.62 | 71.58 | 90.76 | 90.19 | 84.18 | 82.65 |
| MixVal | **71.54** | **52.90** | **78.61** | **65.01** | **67.01** | **72.04** | **92.37** | **92.32** | **85.58** | **83.12** |
| Worst | 15.05 | 4.33 | 15.17 | 16.79 | 12.84 | 71.56 | 90.76 | 88.68 | 83.67 | 80.57 |
| Best | 77.01 | 66.29 | 78.81 | 69.81 | 72.98 | 75.06 | 94.78 | 93.33 | 87.72 | 83.12 |

of practical target-only validation baselines on the large-scale benchmark *DomainNet*. The current state-of-the-art source-based baseline is DEV [18], and for target-only methods, SND [29] holds the state-of-the-art position. Nonetheless, our observations differ from previous findings. When considering the averaged validation accuracy across 6 popular UDA methods, we note that RV consistently outperforms other source-based validation methods across three benchmarks. Regarding target-only validation methods, entropy-based approaches like Entropy and InfoMax prove to be competitive, particularly demonstrating a significant advantage over SND on the *Office-Home* benchmark. As expected, our MixVal consistently attains the highest average accuracy across all four benchmarks, surpassing the vanilla SourceVal baseline, while most existing validation methods may noticeably underperform in comparison to this baseline. Notably, MixVal demonstrates exceptional performance on large-scale benchmarks such as *VisDA* and *DomainNet*, approaching the upper bound of 'Best' and outperforming the second-best method by substantial margins. MixVal's consistent advantages in closed-set UDA highlight its effectiveness in this widely studied setting.

**Partial-set UDA.** Following [29], we assess validation performance in partial-set UDA, a scenario with label shifts between domains. We validate two representative partial-set UDA methods, PADA and SAFN, and present the validation accuracy of *Office-Home* in Table 5. We observe that MixVal consistently outperforms all other validation baselines, maintaining performance close to the 'Best'. However, we notice some differences in comparison to closed-set UDA. Specifically, SND achieves more stable validation performance than Entropy, because Entropy can be susceptible to structure collapse [29]. Second, methods emphasizing 'class diversity', such as InfoMax and Corr-C, often yield lower results than SourceVal. Additionally, RV consistently underperforms SourceVal in scenarios without symmetric label distribution, making it unsuitable for UDA with label shifts.

**Open-partial-set UDA.** We extend our evaluation of UDA scenarios with label shifts, conducting validation for a popular open-partial-set UDA method, DANCE. Table 6 presents the HOS [62, 63] results on *Office-Home*. MixVal consistently provides strong validation performance, nearing the 'Best', while other validation baselines tend to align with the 'Worst'.

**Source-free UDA.** Finally, we compare the target-only validation baselines in source-free UDA. We conduct validation for a popular method, SHOT, and report the accuracy of *Office-31* and *VisDA* in Table 6. We find that MixVal consistently outperforms other baselines in this practical setting.

## 4.3 Analysis

**Influence of pseudo label quality.** Because MixVal utilizes pseudo labels of target data for the *mixup* operation, it is crucial to assess the impact of pseudo label quality. When examining UDA tasks with

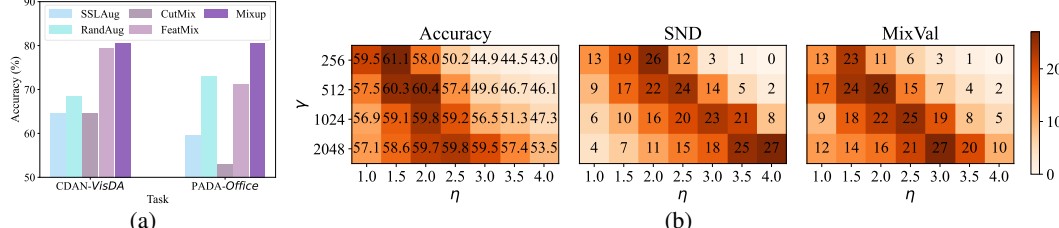

Figure 2: (a) presents the effect of different consistency evaluation techniques. (b) provides comparisons between SND and our MixVal on a validation task involving two hyperparameters: MCC on the Ar→Cl, where $\gamma$ represents the bottleneck dimension and $\eta$ represents the temperature. The SND and MixVal scores of the 28 candidate models are ranked in ascending order. To enhance clarity, we include the real target accuracy for each model in the left figure labeled as 'Accuracy'.

low-quality pseudo labels (around $50\%$ accuracy), such as the results of '→ Cl', it becomes evident that MixVal consistently maintains stable validation performance.

**Influence of consistency evaluation.** MixVal utilizes image-level *mixup*-based consistency evaluation for the target probing. To gain a deeper understanding of its role in MixVal, we substitute it with other consistency evaluation techniques, including instance-based augmentations such as RandAug [64] and SSLAug [65], as well as *mixup* at other levels, such as CutMix [46] and FeatMix [45]. The results in Figure 2(a) consistently demonstrate that image-level *mixup* outperforms other strategies.

Table 7: Two-hyperparameter validation accuracy (%) of closed-set UDA on *Office-Home*.

| Method | MDD [22] | | | | | MCC [25] | | | | |
|---|---|---|---|---|---|---|---|---|---|---|
| | Ar → Cl | Cl → Pr | Pr → Re | Re → Ar | avg | Ar → Cl | Cl → Pr | Pr → Re | Re → Ar | avg |
| SourceVal | 55.99 | 73.15 | 78.77 | 69.39 | 69.33 | 57.91 | 76.84 | 81.13 | 72.89 | 72.19 |
| IWCV [30] | 37.89 | 72.92 | 80.42 | 58.43 | 62.42 | 46.09 | 77.74 | 80.68 | **74.45** | 69.74 |
| DEV [18] | 52.60 | 72.11 | 53.36 | 67.70 | 61.44 | 59.47 | 76.84 | 81.94 | 74.08 | 73.08 |
| RV [31] | **57.59** | 72.25 | **80.83** | 70.79 | 70.37 | 59.13 | 76.84 | 82.03 | 71.98 | 72.50 |
| Entropy [42] | 57.21 | **73.19** | 80.06 | **72.31** | 70.69 | 59.75 | 77.77 | 82.37 | 74.33 | 73.56 |
| InfoMax [28] | **57.59** | 72.92 | 80.06 | **72.31** | **70.72** | 59.70 | **78.73** | **82.58** | 70.33 | 72.84 |
| SND [29] | 38.10 | 56.45 | 70.03 | 65.10 | 57.42 | 53.49 | 74.97 | 77.25 | 74.12 | 69.96 |
| Corr-C [43] | 30.17 | 44.74 | 57.15 | 50.76 | 45.71 | 44.90 | 56.75 | 74.32 | 67.61 | 60.90 |
| MixVal | 55.99 | 72.63 | 80.27 | 72.12 | 70.25 | **60.08** | 78.52 | 81.95 | 74.43 | **73.75** |
| Worst | 30.17 | 39.81 | 53.36 | 50.76 | 43.53 | 43.02 | 56.75 | 73.47 | 67.24 | 60.12 |
| Best | 57.59 | 73.35 | 80.93 | 72.52 | 71.10 | 61.10 | 78.94 | 83.04 | 75.36 | 74.61 |

**Two-hyperparameter validation.** We further evaluate all validation baselines in challenging tasks that involve two hyperparameters. Quantitative comparisons in Table 7 consistently position MixVal as one of the top-performance validators. Qualitative comparisons in Figure 2(b) clearly illustrate that MixVal scores demonstrate a strong correlation with actual accuracy, whereas SND [29] scores exhibit noticeable deviation, resulting in selections far from optimal.

Table 8: Validation accuracy (%) of a mixup-based closed-set UDA method DMRL [50].

| Method | DMRL [50] | | | | *Home* | DMRL [50] | | | *Office* |
|---|---|---|---|---|---|---|---|---|---|
| | →Ar | →Cl | →Pr | →Re | avg | →A | →D | →W | avg |
| Entropy [42] | 58.14 | 50.25 | 69.06 | 71.37 | 62.20 | 63.67 | 80.52 | **86.67** | 76.95 |
| InfoMax [28] | 58.14 | **50.75** | 69.06 | 71.37 | 62.33 | 63.67 | 80.52 | **86.67** | 76.95 |
| SND [29] | 57.74 | 49.96 | **69.28** | 71.42 | 62.10 | 61.84 | **84.14** | **86.67** | 77.55 |
| Corr-C [43] | 58.29 | 49.46 | 68.67 | 71.73 | 62.04 | 60.23 | 77.51 | 81.13 | 72.95 |
| MixVal | **59.13** | 50.41 | **69.28** | 72.07 | **62.72** | **64.89** | 82.93 | **86.67** | **78.16** |
| Worst | 57.71 | 48.99 | 67.78 | 70.72 | 61.30 | 60.23 | 76.31 | 81.13 | 72.55 |
| Best | 59.20 | 50.75 | 69.28 | 72.24 | 62.87 | 65.30 | 84.14 | 86.67 | 78.70 |

**Validation of *mixup*-based UDA method.** The ICE score of MixVal is based on the evaluation of *mixup*-based interpolation consistency during the inference stage. To assess MixVal's robustness, we apply it in the validation of the UDA method DMRL [50], which incorporates *mixup*-based consistency training within the target domain and across two domains at the same time. MixVal is employed to determine the optimal consistency loss coefficient for DMRL, and the results are reported in Table 8. Notably, MixVal consistently outperforms most of the existing target-only validation baselines, even when subjected to the 'attack' posed by specialized UDA model training.

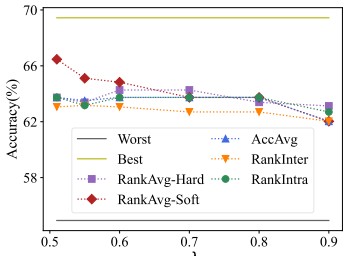

Figure 3: Analysis of probing.

| Method | AdaptSegt [11] | AdvEnt [12] |
|---|---|---|
| **SourceVal** | 39.52 | 39.08 |
| Entropy [42] | 39.47 | 38.41 |
| SND [29] | 40.69 | **40.02** |
| MixVal | **42.20** | **40.02** |
| Worst | 33.84 | 33.06 |
| Best | 42.20 | 41.78 |

Table 9: Segmentation mIoU (%).

| Method | BNM [24] |
|---|---|
| Entropy [42] | 28.21 |
| InfoMax [28] | 28.21 |
| SND [29] | 52.42 |
| Corr-C [43] | 28.21 |
| MixVal | **54.78** |
| Worst | 28.21 |
| Best | 55.16 |

Table 10: ViT results.

**Analysis of probing.** In MixVal, we generate mixed samples by applying *mixup* with a mix ratio $\lambda$ of 0.55 to both target samples and their hard pseudo labels. We then calculate ICE scores for both intra-cluster and inter-cluster mixed samples. The final score for model selection is determined by averaging ascending rankings of both types of ICE scores. Here, we present a comprehensive ablation study to delve into these design choices within MixVal. To do this, we conduct validation experiments for PADA on four partial-set UDA tasks (Ar→Cl, Cl→Pr, Pr→Re, Re→Ar) and report the average accuracy results in Figure 3. 'AccAvg' represents the direct average of ICE scores (i.e., accuracy values) as the final score for model selection. 'RankAvg' denotes the use of the average ranking of ICE scores for selection. 'RankInter' relies solely on inter-cluster mixed samples for probing. 'RankIntra' employs only intra-cluster mixed samples for probing. 'Hard' and 'Soft' indicate the utilization of hard and soft pseudo labels, respectively. It's worth noting that for intra-cluster mixed samples, there is no distinction between 'Hard' and 'Soft' labels, as all pseudo labels involved are the same. The observations drawn from Figure 3 are as follows: (*i*) The mix ratio ($\lambda$) influences the level of ambiguity in mixed samples. In MixVal, $\lambda = 0.55$ generates more ambiguous in-between samples with greater probing capabilities than the easier samples created by $\lambda = 0.9$. Consequently, validation performance at $\lambda = 0.55$ surpasses that at $\lambda = 0.9$. (*ii*) In MixVal, while soft pseudo labels may provide additional performance improvements for inter-cluster probing, the use of hard pseudo labels offers simplicity and stability advantages. (*iii*) The separate probing of each type of mixed samples demonstrates effectiveness, and the combination of intra-cluster and inter-cluster probing enhances MixVal's stability. Kindly refer to Appendix C for more discussions on probing.

**Extension to segmentation.** We extend the use of MixVal to the domain adaptive segmentation task on *GTAV2Cityscapes*, following the same configuration as SND [29]. Our model selection process involves 66 candidate models generated by varying two hyperparameters: the loss coefficient and training iteration. We save checkpoints every 2,000 iterations, starting from the 10,000th iteration to the 30,000th iteration. The results in Table 9 demonstrate that MixVal consistently delivers the top validation performance for both UDA methods, while Entropy consistently underperforms SourceVal.

**Influence of backbone.** In addition to ResNet backbones, we also utilize ViT-B [4] as the backbone for the validation of BNM on the task R→S. Accuracies reported in Table 10 highlight MixVal's superior stability with ViT, especially in comparison to Entropy, InfoMax, and Corr-C.

**Limitations and impacts.** While MixVal excels in classification tasks, further research is needed to address its limitations. This includes establishing the theoretical foundations of probing data structure using pseudo-labeled mixed samples. Additionally, extending MixVal to other machine learning tasks like object detection and regression presents challenges. Poor hyperparameter selections could potentially have negative societal impacts on the real-world deployment of UDA models. Nevertheless, we have not observed such situations during our extensive evaluation of MixVal.

## 5 Conclusion

In summary, we introduce MixVal, a novel and straightforward target-only method for model selection in unsupervised domain adaptation (UDA). MixVal leverages inference-stage *mixup* to generate two distinct types of mixed samples, facilitating effective probing of the learned target structure while elegantly considering key assumptions used in previous approaches. Our extensive evaluations encompass diverse UDA methods and adaptation scenarios, consistently affirming the superior and reliable performance of MixVal when compared to existing model selection methods.

## Acknowledgement

This project is supported by the Singapore Ministry of Education Academic Research Fund Tier 1 (WBS: A-8001229-00-00), a project titled "Towards Robust Single Domain Generalization in Deep Learning". Jian Liang is supported by the National Natural Science Foundation of China (Grant No. 62276256) and the Beijing Nova Program under Grant Z211100002121108.

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

# A Algorithm

The PyTorch-style pseudocode for our validation approach MixVal is presented in Algorithm 1.

---

**Algorithm 1** PyTorch-style pseudocode for MixVal.

---

```python
# x: A batch of real target images with shuffled order.
# lam: The mix ratio, a fixed scalar value between 0.5 and 1.0.
# net: A trained UDA model in the evaluation mode.
# model_list: A list containing candidate UDA models.

# Calculate ICE scores for a mini-batch.
def ice_score(x, lam, net):
    # Random batch index.
    rand_idx = torch.randperm(x.shape[0])
    inputs_a = x
    inputs_b = x[rand_idx]
    # Obtain model predictions and hard pseudo labels.
    pred_a = net(inputs_a)
    pl_a = pred_a.max(dim=1)[1]
    pl_b = pl_a[rand_idx]
    # Intra-cluster mixup.
    same_idx = (pl_a == pl_b).nonzero(as_tuple=True)[0]
    # Inter-cluster mixup.
    diff_idx = (pl_a != pl_b).nonzero(as_tuple=True)[0]
    # Mixup with images and hard pseudo labels.
    mix_inputs = lam * inputs_a + (1 - lam) * inputs_b
    if lam > 0.5:
        mix_labels = pl_a
    else:
        mix_labels = pl_b
    # Obtain predictions for the mixed samples.
    mix_pred = net(mix_inputs)
    mix_pred_labels = mix_pred.max(dim=1)[1]
    # Calculate ICE scores for two-dimensional probing.
    ice_same = torch.sum(mix_pred_labels[same_idx] \
               == mix_labels[same_idx]) / same_idx.shape[0]
    ice_diff = torch.sum(mix_pred_labels[diff_idx] \
               == mix_labels[diff_idx]) / diff_idx.shape[0]
    return ice_same, ice_diff

# Perform model selection based on ICE scores.
def mixVal(model_list, x, lam):
    # Calculate ICE scores for all candidate models.
    ice_same_list = []
    ice_diff_list = []
    for net in model_list:
        ice_same, ice_diff = ice_score(x, lam, net)
        ice_same_list.append(ice_same)
        ice_diff_list.append(ice_diff)
    # Calculate the average rank of two types of ICE scores.
    ice_same_list = torch.tensor(ice_same_list)
    ice_diff_list = torch.tensor(ice_diff_list)
    ice_same_rank = torch.argsort(torch.argsort(ice_same_list))
    ice_diff_rank = torch.argsort(torch.argsort(ice_diff_list))
    average_rank = (ice_same_rank + ice_diff_rank) / 2
    # Choose the model with the highest average rank.
    return model_list[torch.argmax(average_rank)]
```

---

# B  Hyperparameter Settings

In our main experiments, we follow the methodology of previous model selection studies [18, 29]. For simplicity, we tune a single hyperparameter for various UDA methods. In the case of two-hyperparameter validation experiments, as in MCC [25] and MDD [22], we also tune the bottleneck dimension, choosing from the candidate values $\{256, 512, 1024, 2048\}$. This variation is due to observations that the bottleneck dimension varies across different datasets in the official code of these UDA methods. For the validation of semantic segmentation tasks, we also consider two hyperparameters with the training iteration as an additional hyperparameter selected from the set $\{10,000, 12,000, 14,000, 16,000, 18,000, 20,000, 22,000, 24,000, 26,000, 28,000, 30,000\}$, following SND [29]. Detailed hyperparameter settings are available in Table 11.

Table 11: Summary of the considered UDA methods and their corresponding hyperparameters.

| UDA method | UDA Type | Hyperparameter | Search Space | Default Value |
|---|---|---|---|---|
| ATDOC [27] | closed-set self-training | loss coefficient $\lambda$ | $\{0.02, 0.05, 0.1,$ $0.2, 0.5, 1.0, 2.0\}$ | 0.2 |
| BNM [24] | closed-set output regularization | loss coefficient $\lambda$ | $\{0.02, 0.05, 0.1,$ $0.2, 0.5, 1.0, 2.0\}$ | 1.0 |
| CDAN [9] | closed-set feature alignment | loss coefficient $\lambda$ | $\{0.05, 0.1, 0.2,$ $0.5, 1.0, 2.0, 5.0\}$ | 1.0 |
| MCC [25] | closed-set output regularization | temperature $T$ | $\{1.0, 1.5, 2.0,$ $2.5, 3.0, 3.5, 4.0\}$ | 2.5 |
| MDD [22] | closed-set output alignment | margin factor $\gamma$ | $\{0.5, 1.0, 2.0,$ $3.0, 4.0, 5.0, 6.0\}$ | 4.0 |
| SAFN [23] | closed/partial-set feature regularization | loss coefficient $\lambda$ | $\{0.002, 0.005, 0.01,$ $0.02, 0.05, 0.1, 0.2\}$ | 0.05 |
| PADA [15] | partial-set feature alignment | loss coefficient $\lambda$ | $\{0.05, 0.1, 0.2,$ $0.5, 1.0, 2.0, 5.0\}$ | 1.0 |
| DANCE [41] | open-partial-set self-supervision | loss coefficient $\eta$ | $\{0.02, 0.05, 0.1,$ $0.2, 0.5, 1.0, 2.0\}$ | 0.05 |
| SHOT [20] | source-free hypothesis transfer | loss coefficient $\beta$ | $\{0.03, 0.05, 0.1,$ $0.3, 0.5, 1.0, 3.0\}$ | 0.3 |
| DMRL [50] | closed-set *mixup* training | loss coefficient $\lambda$ | $\{0.1, 0.2, 0.5,$ $1.0, 2.0, 5.0, 10.0\}$ | 2.0 |
| AdaptSeg [11] | closed-set output alignment | loss coefficient $\lambda$ | $\{0.0001, 0.0003, 0.001,$ $0.003, 0.01, 0.03\}$ | 0.0002 |
| AdvEnt [12] | closed-set output alignment | loss coefficient $\lambda$ | $\{0.0001, 0.0003, 0.001,$ $0.003, 0.01, 0.03\}$ | 0.001 |

# C  Analysis of Intra-Cluster and Inter-Cluster Probing

In MixVal, we use mixed samples to directly probe the intra-cluster and inter-cluster structures within the target representations learned by a UDA model. This strategy shares similar spirits with the well-known Linear Discriminant Analysis (LDA) [66], which optimizes an objective focused on minimizing intra-cluster variance while maximizing inter-cluster variance. Drawing inspiration from the LDA optimization, we explore a straightforward validation baseline that employs an LDA-like metric. In this context, let's denote the number of target samples as $N$, the number of categories as $K$, and the feature of a sample encoded by a UDA model as $\bar{f}$. The formulation for the LDA score is presented in Equation 1 below.

$$\bar{f} = \frac{1}{N} \sum_{m=1}^{K} \sum_{i=1}^{N_m} f_i^m \qquad \bar{f}_m = \frac{1}{N_m} \sum_{i=1}^{N_m} f_i^m \qquad N = \sum_{m=1}^{K} N_m$$

$$D_{\text{intra}} = \frac{1}{N} \sum_{m=1}^{K} \sum_{i=1}^{N_m} \left\| f_i^m - \bar{f}_m \right\|_2^2 \qquad D_{\text{inter}} = \frac{1}{K} \sum_{m=1}^{K} \left\| \bar{f}_m - \bar{f} \right\|_2^2 \qquad \text{LDA} = \frac{D_{\text{inter}}}{D_{\text{intra}}} \tag{1}$$

In our implementation, we utilize the predicted logits as features and compute variance using the $L_2$ distance. We assess the LDA baseline against all target-only validation methods in both closed-

set UDA (Table 12) and partial-set UDA (Table 13). In both UDA scenarios, LDA outperforms existing validation methods in terms of average validation accuracy. This highlights the advantage of considering two effective assumptions, in contrast to the single assumption used in SND and Entropy. It's worth noting that while LDA performs well, it still falls short of MixVal's performance, underscoring the benefits of MixVal's direct probing approach using mixed samples compared to the indirect probing via measurement of raw predictions used by SND, Entropy, and LDA.

Table 12: Validation accuracy (%) of closed-set UDA on *Office-Home* (*Home*). **bold**: Best value.

| Method | ATDOC [27] | | | | | BNM [24] | | | | | CDAN [9] | | | | |
| | →Ar | →Cl | →Pr | →Re | avg | →Ar | →Cl | →Pr | →Re | avg | →Ar | →Cl | →Pr | →Re | avg |
|---|---|---|---|---|---|---|---|---|---|---|---|---|---|---|---|
| SourceVal | **66.63** | 52.54 | **78.57** | 76.61 | 68.59 | 62.44 | 50.74 | **77.53** | 74.76 | 66.37 | 55.00 | 42.65 | 69.50 | 68.81 | 58.99 |
| Entropy [42] | 63.67 | 55.83 | 76.54 | 78.36 | 68.60 | 66.28 | 54.49 | 74.15 | 77.64 | 68.14 | 67.66 | 57.56 | 76.37 | 77.45 | 69.76 |
| InfoMax [28] | 63.67 | 55.63 | 77.61 | 78.36 | 68.82 | 66.28 | 54.49 | 74.15 | 77.64 | 68.14 | 67.66 | 57.56 | 76.37 | 77.45 | 69.76 |
| SND [29] | 63.67 | 55.63 | 76.54 | 77.54 | 68.34 | 66.28 | 54.49 | 74.15 | 77.64 | 68.14 | **67.94** | 57.56 | **76.96** | 77.68 | **70.04** |
| Corr-C [43] | 63.51 | 50.39 | 73.89 | 73.88 | 65.42 | 58.10 | 45.37 | 68.97 | 70.59 | 60.76 | 53.84 | 41.21 | 64.96 | 67.65 | 56.91 |
| LDA | 63.67 | 55.63 | 76.54 | 77.54 | 68.34 | 66.28 | 54.49 | 74.15 | 77.64 | 68.14 | 67.66 | 57.56 | 76.37 | 77.45 | 69.76 |
| MixVal | 66.47 | **56.87** | 78.14 | **79.20** | **70.17** | **67.36** | **56.18** | 76.10 | **78.12** | **69.44** | 67.71 | **57.78** | 76.89 | **77.76** | 70.03 |
| Worst | 62.89 | 50.39 | 73.89 | 73.88 | 65.26 | 58.10 | 45.37 | 68.96 | 70.59 | 60.75 | 53.80 | 41.21 | 64.78 | 67.65 | 56.86 |
| Best | 68.97 | 58.35 | 80.27 | 80.58 | 72.04 | 68.93 | 57.51 | 78.43 | 79.57 | 71.11 | 68.19 | 57.90 | 77.44 | 78.19 | 70.43 |

| Method | MCC [25] | | | | | MDD [22] | | | | | SAFN [23] | | | | | *Home* |
| | →Ar | →Cl | →Pr | →Re | avg | →Ar | →Cl | →Pr | →Re | avg | →Ar | →Cl | →Pr | →Re | avg | AVG |
|---|---|---|---|---|---|---|---|---|---|---|---|---|---|---|---|---|
| SourceVal | 66.57 | 56.53 | 79.55 | 80.90 | 70.89 | 62.53 | 54.43 | 75.27 | 75.55 | 66.94 | 63.54 | 51.34 | 73.66 | 74.54 | 65.77 | 66.26 |
| Entropy [42] | 69.29 | 59.33 | **80.63** | **80.96** | 72.55 | **66.54** | 57.63 | 77.27 | **77.45** | 69.72 | 59.85 | 46.41 | 72.51 | 73.18 | 62.99 | 68.63 |
| InfoMax [28] | 66.58 | 58.48 | 79.12 | 80.81 | 71.25 | **66.54** | 57.74 | 77.27 | **77.45** | 69.75 | 64.56 | 49.71 | 73.77 | 73.18 | 65.31 | 68.84 |
| SND [29] | 69.05 | 55.61 | 79.72 | 79.10 | 70.87 | 51.34 | 38.01 | **77.61** | 68.46 | 58.86 | 57.90 | 46.41 | 67.04 | 68.18 | 59.88 | 66.02 |
| Corr-C [43] | 69.05 | 55.61 | 79.72 | 79.10 | 70.87 | 47.79 | 31.69 | 63.40 | 60.63 | 50.88 | 62.66 | 46.41 | 68.83 | 68.18 | 61.52 | 61.06 |
| LDA | **70.46** | **59.60** | 80.60 | 80.25 | **72.73** | 66.38 | 57.63 | **77.61** | **77.45** | **69.77** | 64.56 | 49.91 | 72.51 | 71.55 | 64.63 | 68.90 |
| MixVal | 69.79 | 59.24 | 80.47 | 80.74 | 72.56 | 65.73 | **58.01** | 77.36 | 76.91 | 69.50 | **65.98** | **53.14** | **74.76** | **75.40** | **67.32** | **69.84** |
| Worst | 62.72 | 54.63 | 76.19 | 78.19 | 67.93 | 47.79 | 31.69 | 63.40 | 60.63 | 50.88 | 57.90 | 46.41 | 67.04 | 68.18 | 59.88 | 60.26 |
| Best | 70.68 | 59.95 | 80.93 | 81.02 | 73.14 | 66.75 | 58.36 | 77.61 | 77.45 | 70.04 | 66.59 | 53.14 | 74.90 | 75.57 | 67.55 | 70.72 |

Table 13: Validation accuracy (%) of partial-set UDA on *Office-Home*.

| Method | PADA [15] | | | | | SAFN [23] | | | | | *Home* |
| | → Ar | → Cl | → Pr | → Re | avg | → Ar | → Cl | → Pr | → Re | avg | AVG |
|---|---|---|---|---|---|---|---|---|---|---|---|
| SourceVal | 57.21 | 41.90 | 64.48 | 71.89 | 58.87 | 66.82 | 54.71 | 74.41 | 76.48 | 68.11 | 63.49 |
| Entropy [42] | 60.08 | 46.51 | 53.16 | 62.47 | 55.56 | **71.75** | 55.62 | 76.36 | 76.59 | 70.08 | 62.82 |
| InfoMax [28] | 60.08 | **51.40** | 60.20 | 66.67 | 59.59 | 63.67 | 51.74 | 69.64 | 73.62 | 64.67 | 62.13 |
| SND [29] | **67.80** | 50.71 | 59.46 | 67.13 | 61.27 | **71.75** | 51.74 | 76.36 | 78.36 | 69.55 | 65.41 |
| Corr-C [43] | 61.34 | 45.65 | 54.90 | 62.25 | 56.04 | 71.23 | 55.70 | 76.94 | **79.13** | 70.75 | 63.40 |
| LDA | 64.52 | 46.51 | 69.47 | 72.67 | 63.29 | **71.75** | 54.39 | 73.93 | 77.10 | 69.29 | 66.29 |
| MixVal | 67.68 | 51.01 | **72.94** | **78.64** | **67.57** | 71.70 | **57.91** | **77.08** | 78.94 | **71.41** | **69.49** |
| Worst | 56.29 | 39.76 | 50.49 | 59.31 | 51.46 | 62.48 | 49.91 | 68.50 | 73.62 | 63.63 | 57.55 |
| Best | 69.33 | 55.86 | 74.55 | 79.59 | 69.83 | 73.37 | 58.09 | 77.35 | 79.33 | 72.03 | 70.93 |

# D   Full Validation Results

For classification tasks in our evaluation, we utilize the HOS score (%) [62, 63] for open-partial-set UDA and accuracy (%) for all other UDA tasks. In the case of segmentation tasks, we measure mIoU (%) [11, 12]. To accommodate space constraints in this main text, we present the averaged validation results for UDA tasks that share the same target domain. For example, '→ Ar' represents the averaged results of three tasks on the *Office-Home* benchmark, namely 'Cl→ Ar', 'Pr→ Ar', and 'Re→ Ar'. Additionally, please note that the 'avg' row signifies the average of the rows preceding it within each UDA method, while the 'AVG' row represents the average results of 'avg' rows across all considered UDA methods. For more detailed validation results for each specific UDA task, please refer to the corresponding tables, from Table 14 to Table 29.

### Table 14: Accuracy (%) of ATDOC [27], a closed-set UDA method, on *Office-Home*.

| Method | Ar → Cl | Ar → Pr | Ar → Re | Cl → Ar | Cl → Pr | Cl → Re | Pr → Ar | Pr → Cl | Pr → Re | Re → Ar | Re → Cl | Re → Pr | avg |
|---|---|---|---|---|---|---|---|---|---|---|---|---|---|
| SourceVal | 51.43 | **77.31** | 78.17 | **66.87** | 74.36 | 75.60 | 61.85 | 48.06 | 76.06 | 71.16 | 58.14 | **84.05** | 68.59 |
| IWCV [30] | 55.88 | 76.57 | 78.88 | 66.25 | 74.50 | **78.33** | 65.60 | 48.06 | 80.58 | **72.06** | 58.14 | 83.87 | 69.89 |
| DEV [18] | 51.43 | 76.55 | 78.88 | 66.25 | 74.36 | 77.67 | 64.77 | 51.29 | **81.62** | 71.16 | **59.98** | 82.43 | 69.70 |
| RV [31] | 56.38 | 76.12 | **80.01** | 66.25 | 76.80 | **78.33** | **67.82** | **55.62** | 80.58 | 71.98 | 56.40 | 83.87 | **70.85** |
| Entropy [42] | 55.88 | 74.14 | 78.88 | 59.25 | 74.52 | 77.67 | 64.19 | 54.39 | 78.54 | 67.57 | 57.23 | 80.96 | 68.60 |
| InfoMax [28] | 55.88 | 74.14 | 78.88 | 59.25 | 77.74 | 77.67 | 64.19 | 54.39 | 78.54 | 67.57 | 56.61 | 80.96 | 68.82 |
| SND [29] | 55.88 | 74.14 | 78.88 | 59.25 | 74.52 | 75.21 | 64.19 | 54.39 | 78.54 | 67.57 | 56.61 | 80.96 | 68.34 |
| Corr-C [43] | 51.41 | 72.00 | 76.04 | 59.37 | 69.36 | 69.54 | 61.85 | 48.04 | 76.06 | 69.30 | 51.71 | 80.31 | 65.42 |
| MixVal | **57.19** | 74.82 | 79.13 | 64.19 | **78.10** | 77.90 | 65.63 | 55.28 | 80.57 | 69.60 | 58.15 | 81.49 | 70.17 |
| Worst | 51.41 | 72.00 | 76.04 | 59.25 | 69.36 | 69.54 | 61.85 | 48.04 | 76.06 | 67.57 | 51.71 | 80.31 | 65.26 |
| Best | 58.01 | 77.31 | 81.04 | 66.91 | 79.48 | 78.52 | 67.94 | 57.07 | 82.17 | 72.06 | 59.98 | 84.03 | 72.04 |

### Table 15: Accuracy (%) of BNM [24], a closed-set UDA method, on *Office-Home*.

| Method | Ar → Cl | Ar → Pr | Ar → Re | Cl → Ar | Cl → Pr | Cl → Re | Pr → Ar | Pr → Cl | Pr → Re | Re → Ar | Re → Cl | Re → Pr | avg |
|---|---|---|---|---|---|---|---|---|---|---|---|---|---|
| SourceVal | 56.93 | **77.00** | 77.74 | 57.64 | **73.33** | 69.36 | 56.45 | 42.38 | 77.19 | 73.22 | 52.90 | 82.26 | 66.37 |
| IWCV [30] | 46.46 | **77.00** | **79.30** | 63.86 | 61.34 | 62.54 | 63.95 | 42.38 | 78.01 | 71.86 | 55.65 | 83.92 | 65.52 |
| DEV [18] | 57.75 | 71.62 | **79.30** | 57.64 | 67.90 | **75.46** | **66.21** | **54.04** | 78.01 | **73.42** | 57.37 | 82.25 | 68.41 |
| RV [31] | **58.67** | **77.00** | **79.30** | **65.68** | **73.33** | 75.46 | 65.64 | 52.05 | **81.25** | **73.42** | **59.54** | 83.92 | **70.44** |
| Entropy [42] | 53.40 | 67.04 | 78.04 | 63.41 | 71.44 | 73.93 | 63.58 | 52.69 | 80.95 | 71.86 | 57.37 | **83.96** | 68.14 |
| InfoMax [28] | 53.40 | 67.04 | 78.04 | 63.41 | 71.44 | 73.93 | 63.58 | 52.69 | 80.95 | 71.86 | 57.37 | **83.96** | 68.14 |
| SND [29] | 53.40 | 67.04 | 78.04 | 63.41 | 71.44 | 73.93 | 63.58 | 52.69 | 80.95 | 71.86 | 57.37 | **83.96** | 68.14 |
| Corr-C [43] | 46.46 | 67.06 | 74.82 | 49.73 | 61.34 | 62.54 | 56.45 | 42.38 | 74.41 | 68.11 | 47.26 | 78.51 | 60.76 |
| MixVal | 56.30 | 72.90 | 78.75 | 64.66 | 71.44 | 74.52 | 64.32 | **54.04** | 81.09 | 73.09 | 58.19 | **83.96** | 69.44 |
| Worst | 46.46 | 67.04 | 74.82 | 49.73 | 61.34 | 62.54 | 56.45 | 42.38 | 74.41 | 68.11 | 47.26 | 78.51 | 60.75 |
| Best | 58.67 | 77.00 | 80.61 | 67.16 | 74.16 | 76.75 | 66.21 | 54.04 | 81.36 | 73.42 | 59.82 | 84.12 | 71.11 |

### Table 16: Accuracy (%) of CDAN [9], a closed-set UDA method, on *Office-Home*.

| Method | Ar → Cl | Ar → Pr | Ar → Re | Cl → Ar | Cl → Pr | Cl → Re | Pr → Ar | Pr → Cl | Pr → Re | Re → Ar | Re → Cl | Re → Pr | avg |
|---|---|---|---|---|---|---|---|---|---|---|---|---|---|
| SourceVal | 43.41 | 62.51 | 75.51 | 43.96 | 61.59 | 57.70 | 53.75 | 37.53 | 73.22 | 67.28 | 47.01 | 84.39 | 58.99 |
| IWCV [30] | 43.18 | 62.51 | **77.81** | 44.71 | 54.61 | 56.14 | 65.14 | 37.53 | **81.85** | 74.08 | 43.02 | 84.39 | 60.41 |
| DEV [18] | 57.16 | 71.75 | **77.81** | 62.46 | 55.64 | 71.08 | 65.14 | 56.54 | **81.85** | 74.08 | 57.43 | 78.89 | 67.49 |
| RV [31] | 57.16 | 71.75 | 77.78 | 63.62 | 72.92 | 73.40 | 65.14 | 54.50 | **81.85** | **74.21** | 58.56 | 83.37 | 69.52 |
| Entropy [42] | **57.55** | 72.43 | 77.74 | 63.62 | 72.92 | 73.40 | **65.27** | **56.66** | 81.20 | 74.08 | 58.47 | 83.76 | 69.76 |
| InfoMax [28] | **57.55** | 72.43 | 77.74 | 63.62 | 72.92 | 73.40 | **65.27** | **56.66** | 81.20 | 74.08 | 58.47 | 83.76 | 69.76 |
| SND [29] | **57.55** | 72.43 | 77.78 | **64.61** | **73.73** | 73.40 | 65.14 | **56.66** | **81.85** | 74.08 | 58.47 | **84.73** | 70.04 |
| Corr-C [43] | 43.14 | 63.05 | 73.61 | 43.96 | 54.58 | 56.12 | 51.75 | 37.50 | 73.22 | 65.80 | 43.00 | 77.25 | 56.91 |
| MixVal | **57.55** | **73.71** | 77.75 | 63.95 | 73.19 | **74.06** | 65.23 | 56.59 | 81.47 | 73.95 | **59.19** | 83.76 | 70.03 |
| Worst | 43.14 | 62.51 | 73.61 | 43.96 | 54.58 | 56.12 | 51.63 | 37.50 | 73.22 | 65.80 | 43.00 | 77.25 | 56.86 |
| Best | 57.55 | 73.71 | 78.33 | 64.61 | 73.89 | 74.39 | 65.76 | 56.66 | 81.85 | 74.21 | 59.50 | 84.73 | 70.43 |

### Table 17: Accuracy (%) of MCC [25], a closed-set UDA method, on *Office-Home*.

| Method | Ar → Cl | Ar → Pr | Ar → Re | Cl → Ar | Cl → Pr | Cl → Re | Pr → Ar | Pr → Cl | Pr → Re | Re → Ar | Re → Cl | Re → Pr | avg |
|---|---|---|---|---|---|---|---|---|---|---|---|---|---|
| SourceVal | 57.23 | 78.19 | **81.75** | 60.65 | 76.50 | **78.79** | 64.15 | 53.15 | 82.17 | **74.91** | 59.20 | 83.96 | 70.89 |
| IWCV [30] | **60.02** | 78.15 | 81.34 | 68.73 | **78.51** | 77.85 | 64.15 | **57.85** | 81.04 | 73.18 | 58.92 | 84.46 | 72.02 |
| DEV [18] | 57.16 | 78.15 | 81.34 | **69.10** | 73.01 | 76.80 | 64.15 | **57.85** | 82.17 | **74.91** | 59.20 | 84.46 | 71.38 |
| RV [31] | 59.34 | **78.53** | 80.70 | **69.10** | 77.83 | 78.22 | 67.20 | **57.85** | 82.24 | **74.91** | 59.20 | 85.54 | **72.56** |
| Entropy [42] | 59.31 | **78.53** | 81.59 | 66.87 | 77.83 | **78.79** | 67.20 | **57.85** | 82.51 | 73.79 | 60.82 | **85.54** | 72.55 |
| InfoMax [28] | **60.02** | 74.66 | **81.75** | 64.98 | 78.24 | 78.49 | 64.15 | 54.52 | 82.19 | 70.62 | 60.89 | 84.46 | 71.25 |
| SND [29] | 53.56 | 77.43 | 79.46 | 67.28 | 76.48 | 76.80 | 65.06 | 54.34 | 81.04 | 74.82 | 58.92 | 85.24 | 70.87 |
| Corr-C [43] | 53.56 | 77.43 | 79.46 | 67.28 | 76.48 | 76.80 | 65.06 | 54.34 | 81.04 | 74.82 | 58.92 | 85.24 | 70.87 |
| MixVal | 59.08 | 77.81 | 81.61 | 68.40 | 78.45 | 78.40 | **67.56** | 57.65 | 82.21 | 73.41 | **61.00** | 85.15 | **72.56** |
| Worst | 53.56 | 73.44 | 79.25 | 60.65 | 73.01 | 75.76 | 59.74 | 53.15 | 79.55 | 67.78 | 57.18 | 82.11 | 67.93 |
| Best | 60.02 | 78.53 | 81.75 | 69.22 | 78.51 | 78.79 | 67.90 | 58.49 | 82.51 | 74.91 | 61.35 | 85.74 | 73.14 |

### Table 18: Accuracy (%) of MDD [22], a closed-set UDA method, on *Office-Home*.

| Method | Ar → Cl | Ar → Pr | Ar → Re | Cl → Ar | Cl → Pr | Cl → Re | Pr → Ar | Pr → Cl | Pr → Re | Re → Ar | Re → Cl | Re → Pr | avg |
|---|---|---|---|---|---|---|---|---|---|---|---|---|---|
| SourceVal | 54.85 | 73.35 | 77.05 | 58.76 | 69.95 | 72.23 | 60.03 | 51.02 | 77.36 | 68.81 | 57.42 | 82.50 | 66.94 |
| IWCV [30] | 56.40 | 69.52 | 76.59 | 58.76 | 67.40 | 69.43 | 61.89 | 56.43 | 76.82 | 71.94 | 56.68 | **84.43** | 67.19 |
| DEV [18] | 57.71 | **75.42** | 77.05 | 58.76 | **72.99** | 70.51 | **63.95** | 56.43 | 80.26 | 70.54 | 56.68 | 82.14 | 68.54 |
| RV [31] | **58.05** | **75.42** | 76.59 | 63.54 | 69.95 | **73.74** | **63.95** | 51.02 | **80.38** | **72.23** | 58.17 | **84.43** | 68.96 |
| Entropy [42] | 57.73 | 74.54 | **78.22** | **64.07** | **72.99** | **73.74** | **63.95** | 55.85 | **80.38** | 71.61 | 59.31 | 84.28 | 69.72 |
| InfoMax [28] | **58.05** | 74.54 | **78.22** | **64.07** | **72.99** | **73.74** | **63.95** | 55.85 | **80.38** | 71.61 | 59.31 | 84.28 | **69.75** |
| SND [29] | **58.05** | **75.42** | 77.05 | 44.99 | **72.99** | 48.06 | 37.08 | 21.60 | 80.26 | 71.94 | 34.39 | **84.43** | 58.86 |
| Corr-C [43] | 39.08 | 59.74 | 69.61 | 44.99 | 54.58 | 48.06 | 37.08 | 21.60 | 64.22 | 61.31 | 34.39 | 75.87 | 50.88 |
| MixVal | 57.39 | 75.13 | 78.15 | 63.45 | 72.67 | 72.69 | 63.91 | **56.63** | 79.90 | 69.82 | **60.02** | 84.27 | 69.50 |
| Worst | 39.08 | 59.74 | 69.61 | 44.99 | 54.58 | 48.06 | 37.08 | 21.60 | 64.22 | 61.31 | 34.39 | 75.87 | 50.88 |
| Best | 58.05 | 75.42 | 78.22 | 64.07 | 72.99 | 73.74 | 63.95 | 57.02 | 80.38 | 72.23 | 60.02 | 84.43 | 70.04 |

Table 19: Accuracy (%) of SAFN [23], a closed-set UDA method, on *Office-Home*.

| Method | Ar→Cl | Ar→Pr | Ar→Re | Cl→Ar | Cl→Pr | Cl→Re | Pr→Ar | Pr→Cl | Pr→Re | Re→Ar | Re→Cl | Re→Pr | avg |
|---|---|---|---|---|---|---|---|---|---|---|---|---|---|
| SourceVal | 50.78 | 69.72 | 76.06 | 59.66 | 70.29 | 69.86 | 60.90 | 46.07 | **77.71** | 70.05 | **57.16** | 80.96 | 65.77 |
| IWCV [30] | 50.24 | 69.72 | **77.28** | 62.63 | 67.24 | 69.86 | 58.84 | 49.69 | 75.72 | **71.45** | **57.16** | 79.97 | 65.82 |
| DEV [18] | 51.07 | 69.72 | 76.64 | 59.66 | 67.24 | 71.26 | 58.84 | 49.69 | 75.72 | 70.95 | 50.65 | 76.64 | 64.84 |
| RV [31] | 51.07 | 71.41 | 76.64 | 62.63 | 68.44 | 70.44 | 58.84 | 44.49 | **77.71** | **71.45** | 54.82 | **81.46** | 65.78 |
| Entropy [42] | 45.93 | 69.72 | 75.49 | 55.29 | 67.22 | 68.35 | 54.26 | 43.30 | 75.69 | 70.00 | 49.99 | 80.60 | 62.99 |
| InfoMax [28] | 50.47 | 69.72 | 75.49 | 62.46 | **70.98** | 68.35 | 61.23 | 43.30 | 75.69 | 70.00 | 55.37 | 80.60 | 65.31 |
| SND [29] | 45.93 | 64.36 | 70.60 | 55.29 | 60.13 | 62.50 | 54.26 | 43.30 | 71.43 | 64.15 | 49.99 | 76.64 | 59.88 |
| Corr-C [43] | 45.93 | 69.72 | 70.60 | 55.29 | 60.13 | 62.50 | 61.23 | 43.30 | 71.43 | **71.45** | 49.99 | 76.64 | 61.52 |
| MixVal | **51.73** | **72.14** | 77.08 | **63.92** | **70.98** | **71.55** | **62.71** | **50.52** | 77.56 | 71.30 | **57.16** | 81.17 | **67.32** |
| Worst | 45.93 | 64.36 | 70.60 | 55.29 | 60.13 | 62.50 | 54.26 | 43.30 | 71.43 | 64.15 | 49.99 | 76.64 | 59.88 |
| Best | 51.73 | 72.27 | 77.30 | 64.65 | 70.98 | 71.70 | 63.66 | 50.52 | 77.71 | 71.45 | 57.16 | 81.46 | 67.55 |

Table 20: Accuracy (%) of closed-set UDA methods on *Office-31*.

| Method | ATDOC [27] | | | | | BNM [24] | | | | | CDAN [9] | | | | |
|---|---|---|---|---|---|---|---|---|---|---|---|---|---|---|---|
| | A→D | A→W | D→A | W→A | avg | A→D | A→W | D→A | W→A | avg | A→D | A→W | D→A | W→A | avg |
| SourceVal | 88.96 | **87.80** | 73.65 | 71.46 | 80.47 | **90.36** | **89.43** | 73.13 | 72.70 | **81.41** | 91.16 | **89.06** | 66.33 | 61.46 | 77.00 |
| IWCV [30] | 86.14 | 86.54 | 73.65 | 71.46 | 79.45 | 85.54 | **89.43** | 73.13 | 72.70 | 80.20 | 69.08 | 58.74 | 66.33 | 61.46 | 63.90 |
| DEV [18] | 86.14 | 86.54 | 73.65 | 71.46 | 79.45 | 85.54 | **89.43** | 73.13 | 72.70 | 80.20 | 91.16 | 88.30 | 66.33 | 61.46 | 76.81 |
| RV [31] | 89.96 | 87.23 | **74.28** | **75.58** | **81.76** | 88.55 | **89.43** | 74.90 | 66.52 | 79.85 | 91.16 | 88.30 | **76.18** | 70.36 | 81.50 |
| Entropy [42] | 86.14 | **87.80** | 73.87 | 72.70 | 80.13 | 85.54 | 83.14 | 71.07 | 74.26 | 78.50 | 91.16 | **89.06** | 72.88 | 70.36 | 80.87 |
| InfoMax [28] | 86.14 | **87.80** | 73.87 | 72.70 | 80.13 | 85.54 | 83.14 | 71.07 | 69.97 | 77.43 | 91.16 | 88.30 | 72.88 | 70.36 | 80.68 |
| SND [29] | **92.37** | **87.80** | 73.87 | 72.70 | 81.69 | 85.54 | 83.14 | 74.62 | 74.26 | 79.39 | 92.37 | 88.55 | 72.88 | 70.22 | 81.01 |
| Corr-C [43] | 90.96 | 84.40 | 71.88 | 70.22 | 79.37 | 84.34 | 78.99 | 67.80 | 66.52 | 74.41 | 67.67 | 59.62 | 58.15 | 58.43 | 60.97 |
| MixVal | 90.96 | 86.54 | 73.75 | 73.47 | 81.18 | 86.48 | 87.00 | **75.64** | **74.29** | 80.85 | **92.64** | **89.06** | 75.08 | **70.38** | **81.79** |
| Worst | 86.14 | 84.40 | 71.88 | 70.22 | 78.16 | 84.34 | 78.99 | 67.80 | 66.52 | 74.41 | 67.67 | 57.11 | 58.15 | 58.43 | 60.34 |
| Best | 92.37 | 87.80 | 75.04 | 75.58 | 82.70 | 90.36 | 89.43 | 75.75 | 75.29 | 82.71 | 92.77 | 89.06 | 76.18 | 70.57 | 82.15 |

Table 21: Accuracy (%) of closed-set UDA methods on *Office-31*.

| Method | MCC [25] | | | | | MDD [22] | | | | | SAFN [23] | | | | |
|---|---|---|---|---|---|---|---|---|---|---|---|---|---|---|---|
| | A→D | A→W | D→A | W→A | avg | A→D | A→W | D→A | W→A | avg | A→D | A→W | D→A | W→A | avg |
| SourceVal | 90.96 | 91.07 | 73.33 | 72.89 | 82.06 | 91.06 | 86.23 | 76.68 | 74.76 | 82.18 | 83.73 | 87.17 | 68.96 | 69.44 | 77.33 |
| IWCV [30] | 91.16 | 88.55 | 73.33 | 72.89 | 81.48 | 91.16 | 89.18 | 76.68 | 74.30 | 82.83 | 86.55 | 80.38 | 68.96 | **69.68** | 76.39 |
| DEV [18] | 89.16 | 93.08 | 73.33 | 72.06 | 81.91 | 91.16 | 89.18 | 76.68 | 74.62 | 82.91 | 86.55 | 80.38 | 68.96 | 67.45 | 75.84 |
| RV [31] | 89.06 | 93.08 | 74.42 | 73.52 | 82.52 | **92.57** | 86.79 | 73.94 | 74.97 | 82.07 | 90.83 | 87.17 | 68.76 | 68.62 | 78.85 |
| Entropy [42] | 90.56 | **93.46** | 74.83 | 73.02 | 82.97 | **92.57** | 90.82 | **78.03** | 74.58 | 84.00 | **91.57** | 85.66 | 67.20 | 69.26 | 78.42 |
| InfoMax [28] | 89.16 | 88.55 | 74.16 | **73.70** | 81.39 | **92.57** | 90.82 | **78.03** | 74.97 | **84.10** | **91.57** | **87.42** | 67.20 | 69.26 | 78.86 |
| SND [29] | **91.97** | **93.46** | 74.83 | 73.02 | **83.32** | 92.17 | 90.82 | **78.03** | 74.97 | 84.00 | 89.96 | 85.66 | 67.20 | 69.26 | 78.02 |
| Corr-C [43] | 91.37 | **93.46** | 74.83 | 73.02 | 83.17 | 91.57 | 85.66 | 73.91 | 74.58 | 81.43 | 86.75 | 80.38 | 67.09 | **69.68** | 75.98 |
| MixVal | 91.77 | 93.21 | 74.74 | 73.44 | 83.29 | 91.77 | **91.74** | 77.35 | 74.58 | 83.86 | 89.96 | 86.83 | **69.91** | 69.31 | **79.00** |
| Worst | 86.75 | 87.17 | 71.18 | 69.93 | 78.76 | 87.35 | 85.66 | 73.91 | 72.20 | 79.78 | 83.73 | 80.38 | 67.09 | 67.45 | 74.66 |
| Best | 91.97 | 93.46 | 74.83 | 74.01 | 83.57 | 92.57 | 92.20 | 78.03 | 75.01 | 84.45 | 91.57 | 87.42 | 70.43 | 69.68 | 79.78 |

Table 22: Accuracy (%) of CDAN [9], a closed-set UDA method, on *DomainNet*.

| Method | C→S | P→C | P→R | R→C | R→P | R→S | S→P | avg |
|---|---|---|---|---|---|---|---|---|
| Entropy [42] | **58.04** | **64.78** | **74.42** | **69.39** | **68.65** | **60.63** | 62.94 | **65.55** |
| InfoMax [28] | **58.04** | **64.78** | **74.42** | **69.39** | **68.65** | **60.63** | 62.94 | **65.55** |
| SND [29] | **58.04** | **64.78** | **74.42** | **69.39** | **68.65** | **60.63** | 60.70 | 65.23 |
| Corr-C [43] | **58.04** | 57.73 | **74.42** | 56.98 | 65.07 | 51.23 | 60.70 | 60.60 |
| MixVal | **58.04** | **64.78** | **74.42** | **69.39** | **68.65** | **60.63** | 62.94 | **65.55** |
| Worst | 51.59 | 57.73 | 73.44 | 56.98 | 63.06 | 51.23 | 58.46 | 58.93 |
| Best | 58.04 | 64.78 | 74.44 | 69.39 | 68.65 | 60.63 | 62.94 | 65.55 |

Table 23: Accuracy (%) of BNM [24], a closed-set UDA method, on *DomainNet*.

| Method | C→S | P→C | P→R | R→C | R→P | R→S | S→P | avg |
|---|---|---|---|---|---|---|---|---|
| Entropy [42] | 56.42 | 61.57 | 74.31 | 65.15 | 65.15 | 40.95 | 63.42 | 61.00 |
| InfoMax [28] | 56.42 | 68.95 | 74.31 | 65.15 | 65.15 | 54.93 | 63.42 | 64.05 |
| SND [29] | 43.78 | 61.57 | 74.31 | 51.55 | 54.40 | 40.95 | 54.59 | 54.45 |
| Corr-C [43] | 43.78 | 60.03 | 77.62 | 59.47 | 67.19 | 40.95 | 59.64 | 58.38 |
| MixVal | **58.48** | **69.63** | **78.68** | **66.05** | **67.59** | **58.50** | **65.20** | **66.30** |
| Worst | 43.78 | 60.03 | 74.31 | 51.55 | 54.40 | 40.95 | 54.59 | 54.23 |
| Best | 58.48 | 69.63 | 78.68 | 66.10 | 67.79 | 58.50 | 65.20 | 66.34 |

Table 24: Accuracy (%) of ATDOC [27], a closed-set UDA method, on *DomainNet*.

| Method | C→S | P→C | P→R | R→C | R→P | R→S | S→P | avg |
|---|---|---|---|---|---|---|---|---|
| Entropy [42] | 46.43 | 65.98 | **79.60** | 61.52 | 64.24 | 57.92 | 59.46 | 62.16 |
| InfoMax [28] | 46.43 | 65.98 | **79.60** | 61.52 | 64.24 | 57.92 | 59.46 | 62.16 |
| SND [29] | 46.43 | 65.98 | **79.60** | 61.52 | 64.24 | 47.58 | 59.46 | 60.69 |
| Corr-C [43] | 54.71 | 60.63 | 74.42 | 59.33 | 64.58 | 52.66 | 59.95 | 60.90 |
| MixVal | **62.11** | **69.64** | **79.60** | **68.24** | **69.79** | **61.35** | **67.10** | **68.26** |
| Worst | 46.43 | 60.63 | 74.42 | 59.33 | 64.24 | 47.58 | 59.46 | 58.87 |
| Best | 63.12 | 71.14 | 80.38 | 69.45 | 69.79 | 61.35 | 67.10 | 68.26 |

Table 25: Accuracy (%) of PADA [15], a partial-set UDA method, on *Office-Home*.

| Method | Ar→Cl | Ar→Pr | Ar→Re | Cl→Ar | Cl→Pr | Cl→Re | Pr→Ar | Pr→Cl | Pr→Re | Re→Ar | Re→Cl | Re→Pr | avg |
|---|---|---|---|---|---|---|---|---|---|---|---|---|---|
| SourceVal | 45.03 | 68.85 | 81.89 | 43.25 | 46.83 | 57.26 | 57.12 | 36.42 | 76.53 | 71.26 | 44.24 | 77.76 | 58.87 |
| IWCV [30] | **55.58** | 65.10 | **84.54** | 51.42 | **61.29** | 53.01 | 57.02 | 35.16 | 81.34 | 70.52 | **60.78** | 74.12 | 62.49 |
| DEV [18] | 54.81 | **78.15** | 78.02 | **58.13** | **61.29** | 50.14 | 67.86 | 35.16 | 83.21 | 74.66 | 57.91 | 77.76 | 64.76 |
| RV [31] | 43.22 | 65.10 | 81.89 | 42.70 | 48.74 | 52.79 | 57.21 | 35.16 | 77.80 | 73.46 | 44.24 | 77.76 | 58.34 |
| Entropy [42] | 40.12 | 40.11 | 55.94 | 52.43 | 37.25 | 50.14 | 57.30 | 47.22 | 81.34 | 70.52 | 52.18 | 82.13 | 55.56 |
| InfoMax [28] | 54.81 | 69.24 | 78.02 | 52.43 | 37.25 | 50.14 | 57.30 | 47.22 | 71.84 | 70.52 | 52.18 | 74.12 | 59.59 |
| SND [29] | 40.12 | 40.11 | 55.94 | **58.13** | 56.13 | 64.11 | 70.62 | **51.22** | 81.34 | 74.66 | **60.78** | 82.13 | 61.27 |
| Corr-C [43] | 40.12 | 40.11 | 55.94 | 54.18 | 46.89 | 53.01 | 58.59 | 38.93 | 77.80 | 71.26 | 57.91 | 77.70 | 56.04 |
| MixVal | 45.02 | **78.15** | 83.69 | 56.23 | 57.85 | **68.19** | **71.41** | 47.22 | **84.04** | 75.39 | **60.78** | **82.82** | 67.57 |
| Worst | 40.12 | 40.11 | 55.94 | 41.41 | 37.25 | 50.14 | 56.93 | 34.87 | 71.84 | 70.52 | 44.30 | 74.12 | 51.46 |
| Best | 55.58 | 78.15 | 86.53 | 58.13 | 61.29 | 68.19 | 73.00 | 51.22 | 84.04 | 76.86 | 60.78 | 84.20 | 69.83 |

Table 26: Accuracy (%) of SAFN [23], a partial-set UDA method, on *Office-Home*.

| Method | Ar→Cl | Ar→Pr | Ar→Re | Cl→Ar | Cl→Pr | Cl→Re | Pr→Ar | Pr→Cl | Pr→Re | Re→Ar | Re→Cl | Re→Pr | avg |
|---|---|---|---|---|---|---|---|---|---|---|---|---|---|
| SourceVal | **59.40** | 77.14 | 81.34 | 63.97 | 67.00 | 71.29 | 65.60 | 46.21 | 76.81 | 70.89 | 58.51 | 79.10 | 68.11 |
| IWCV [30] | 52.24 | 74.45 | **82.16** | **70.98** | 62.41 | 70.18 | 63.45 | 53.49 | 76.81 | 73.65 | 56.00 | 78.49 | 67.86 |
| DEV [18] | 55.22 | 74.45 | 80.07 | **70.98** | 67.00 | 71.29 | 63.45 | 51.70 | 76.81 | 73.65 | 57.91 | 80.39 | 68.58 |
| RV [31] | 53.67 | 71.60 | 81.34 | 67.58 | 67.00 | 73.27 | 65.70 | 48.54 | 76.81 | 73.65 | 56.00 | 79.89 | 67.92 |
| Entropy [42] | 58.93 | 74.90 | 80.73 | **70.98** | 74.12 | 69.80 | 70.16 | 50.09 | 79.24 | 74.10 | 57.85 | 80.06 | 70.08 |
| InfoMax [28] | 51.82 | 67.62 | 76.97 | 64.65 | 65.77 | 69.80 | 59.69 | 50.09 | 74.10 | 66.67 | 53.31 | 75.52 | 64.67 |
| SND [29] | 51.82 | 74.90 | 80.73 | **70.98** | **74.12** | **75.10** | 70.16 | 50.09 | 79.24 | 74.10 | 53.08 | 80.06 | 69.55 |
| Corr-C [43] | **59.40** | **77.20** | **82.16** | 67.58 | 72.89 | **75.10** | 70.16 | **55.70** | 80.12 | 75.94 | 52.00 | **80.73** | 70.75 |
| MixVal | 59.24 | 76.90 | 81.28 | 68.96 | 73.71 | 74.82 | **70.19** | 55.34 | **80.73** | 75.94 | **59.16** | 80.62 | **71.41** |
| Worst | 51.52 | 67.62 | 76.97 | 61.07 | 62.35 | 69.80 | 59.69 | 46.21 | 74.10 | 66.67 | 52.00 | 75.52 | 63.63 |
| Best | 59.40 | 77.20 | 82.16 | 71.72 | 74.12 | 75.10 | 72.45 | 55.70 | 80.73 | 75.94 | 59.16 | 80.73 | 72.03 |

Table 27: HOS [62, 63] (%) of DANCE [41], an open-partial-set UDA method, on *Office-Home*.

| Method | Ar→Cl | Ar→Pr | Ar→Re | Cl→Ar | Cl→Pr | Cl→Re | Pr→Ar | Pr→Cl | Pr→Re | Re→Ar | Re→Cl | Re→Pr | avg |
|---|---|---|---|---|---|---|---|---|---|---|---|---|---|
| Entropy [42] | 38.29 | 26.08 | 36.51 | 32.92 | 17.10 | 32.19 | 37.69 | 46.40 | 45.53 | 25.39 | 33.75 | 39.37 | 34.27 |
| InfoMax [28] | 38.29 | 26.08 | 36.51 | 32.92 | 17.10 | 32.19 | 37.69 | 46.40 | 45.33 | 25.39 | 33.75 | 39.37 | 34.25 |
| SND [29] | 1.00 | 0.00 | 12.73 | 0.00 | 42.84 | 1.95 | 19.77 | 11.99 | 35.69 | 25.39 | 0.00 | 28.40 | 14.98 |
| Corr-C [43] | 1.00 | 0.00 | 12.73 | 0.00 | 42.84 | 1.95 | 19.77 | 11.99 | 35.69 | 25.39 | 69.02 | 28.40 | 18.62 |
| MixVal | **47.93** | **76.36** | 66.57 | **67.87** | **75.17** | 59.05 | 69.18 | 58.93 | 69.40 | 77.57 | 51.83 | 84.31 | 67.01 |
| Worst | 1.00 | 0.00 | 12.73 | 0.00 | 17.10 | 1.95 | 19.77 | 11.99 | 35.69 | 25.39 | 0.00 | 28.40 | 12.84 |
| Best | 67.00 | 76.96 | 66.57 | 71.76 | 75.17 | 69.99 | 77.42 | 64.32 | 72.87 | 81.84 | 67.54 | 84.31 | 72.98 |

Table 28: Accuracy (%) of DMRL [50], a closed-set UDA method, on *Office-Home*.

| Method | Ar→Cl | Ar→Pr | Ar→Re | Cl→Ar | Cl→Pr | Cl→Re | Pr→Ar | Pr→Cl | Pr→Re | Re→Ar | Re→Cl | Re→Pr | avg |
|---|---|---|---|---|---|---|---|---|---|---|---|---|---|
| Entropy [42] | 47.03 | 62.09 | 73.42 | 55.29 | **64.34** | 66.54 | 51.92 | **48.96** | 74.16 | 67.20 | **54.75** | **80.74** | 62.20 |
| InfoMax [28] | **48.55** | 62.09 | 73.42 | 55.29 | **64.34** | 66.54 | 51.92 | **48.96** | 74.16 | 67.20 | **54.75** | **80.74** | 62.33 |
| SND [29] | 47.84 | **62.76** | **73.86** | 54.92 | **64.34** | 66.65 | 51.87 | **48.96** | 73.74 | 66.42 | 53.08 | **80.74** | 62.10 |
| Corr-C [43] | 47.74 | **62.76** | 72.53 | 55.29 | **64.34** | **68.12** | 53.11 | 47.56 | 74.55 | 66.46 | 53.08 | 78.91 | 62.04 |
| MixVal | 47.74 | **62.76** | 73.42 | **56.29** | **64.34** | **68.12** | 53.73 | **48.96** | 74.67 | 67.37 | 54.54 | **80.74** | **62.72** |
| Worst | 47.03 | 61.73 | 72.34 | 54.84 | 62.90 | 66.15 | 51.87 | 46.85 | 73.67 | 66.42 | 53.08 | 78.71 | 61.30 |
| Best | 48.55 | 62.76 | 73.86 | 56.41 | 64.34 | 68.12 | 53.73 | 48.96 | 74.73 | 67.45 | 54.75 | 80.74 | 62.87 |

Table 29: Accuracy (%) of DMRL [50] and SHOT [20] on *Office-31*.

| Method | DMRL [50] | | | | | SHOT [20] | | | | |
|---|---|---|---|---|---|---|---|---|---|---|
| | A→D | A→W | D→A | W→A | avg | A→D | A→W | D→A | W→A | avg |
| Entropy [42] | 80.52 | **86.67** | 61.77 | 65.57 | 73.63 | 90.76 | 88.68 | **71.21** | 72.13 | 80.70 |
| InfoMax [28] | 80.52 | **86.67** | 61.77 | 65.57 | 73.63 | 90.76 | 88.68 | **71.21** | 72.13 | 80.70 |
| SND [29] | **84.14** | **86.67** | 61.77 | 61.91 | 73.62 | 90.76 | 88.68 | **71.21** | 72.13 | 80.70 |
| Corr-C [43] | 77.51 | 81.13 | 60.28 | 60.17 | 69.77 | 90.76 | 90.19 | **71.21** | 71.96 | 81.03 |
| MixVal | 82.93 | **86.67** | **63.40** | **66.38** | **74.85** | **92.37** | **92.32** | **71.21** | **72.88** | **82.20** |
| Worst | 76.31 | 81.13 | 60.28 | 60.17 | 69.47 | 90.76 | 88.68 | 71.21 | 71.92 | 80.64 |
| Best | 84.14 | 86.67 | 64.22 | 66.38 | 75.35 | 94.78 | 93.33 | 75.58 | 74.55 | 84.56 |

