# OpenReview forum: "Mixed Samples as Probes for Unsupervised Model Selection in Domain Adaptation"
_NeurIPS.cc/2023/Conference — NeurIPS 2023 poster_

### Official Review · Reviewer_zt6b · 2023-07-05

**Soundness:** 3 good
**Presentation:** 3 good
**Contribution:** 4 excellent
**Rating:** 7
**Confidence:** 3

**Summary:**

This paper proposed a new framework to validate the domain-adapted model through Mixed samples. Leveraging mixed samples is not new, but application to the domain-adapted model validation is novel. The framework is very general, so it can be combined with any adaptation methods. Experiments are very extensive over various domain adaptation methods over different validation methods including SND and Corr-C, and the authors provide the supporting evidence as much as possible. In many cases, the proposed framework shows superior performance.

**Strengths:**

In the domain adaptation scenario, the target labels are generally not available. The performance of the adapted model is not typically very robust, so we might want to try the adaptation several times, but we do not know which model is better. So a good model validation framework is necessary. Therefore, the problem is well-defined and well-motivated in practice.

The main strengths of this paper are 1) simplicity, 2) generalizability, and 3) superior performance.
In addition, the claimed arguments are well-supported by extensive experiments, including Segmentation and Backbone change L301-304.

**Weaknesses:**

The main weakness of this paper is (as the authors also mentioned) the lack of theoretical analysis. This paper would add more value if the authors provided meaningful analysis to support the framework.

**Questions:**

1. L194-195: How can we calculate ICE score as an accuracy between $y^{i}$ and $\hat{y}^{i}=(...,1/2,...,1/2,...)$ (for example) because $\hat{y}$ is one-hot pseudo-label according to L189?
2. L244: Any more justification on why $\lambda=0.55$? How's the performance variation depending on the choice of $\lambda$?

**Limitations:**

I'm on the positive side, but without looking at the codes and theoretical justifications, the reproducibility cannot be fully verified.

---

> ### Author Rebuttal · Authors · 2023-08-10
>
> We are sincerely grateful for both your recognition of our contributions and your valuable and comprehensive feedback. We've taken each of your concerns into account and have provided detailed responses below.
>
> > **Q1**:  The main weakness of this paper is (as the authors also mentioned) the lack of theoretical analysis. This paper would add more value if the authors provided meaningful analysis to support the framework.
>
> **A1**: Thank you for your constructive suggestion. We acknowledge rigorous theoretical analysis of our method could shed more light on the challenge of model selection in domain adaptation. We recognize this as an avenue for future work. Although our current submission lacks extensive theoretical analysis, we have diligently compensated by offering an in-depth empirical analysis of our validation method, MixVal. This analysis is further elaborated in Appendix B and is accompanied by new comparisons, including **a comparison with the combination of Entropy [44] and SND [27], as depicted in Table 1 of the attached PDF in the global rebuttal.**
>
> > **Q2**:  L194-195: How can we calculate ICE score as an accuracy between $y^i$ and $y^i = (...,1/2,...,1/2,...) $ (for example) because
>  $\hat{y}$ is one-hot pseudo-label according to L189?
>
> **A2**: The notation $\hat{y}$ denotes the pseudo label predicted by the fixed UDA model. In particular, $\hat{y}_t^i$ signifies the pseudo label for target sample $x_t^i$. During mixup, when we combine target samples $x_t^i$ and $x_t^j$ to create a mixed sample, we utilize the hard one-hot labels from $\hat{y}_t^i$ and $\hat{y}_t^j$ along with a mix ratio exceeding 0.5. This choice ensures that the interpolated label of the mixed sample **avoids the scenario of (...,1/2,...,1/2,...)**. When computing the ICE score for this mixed sample, we predict its hard pseudo label using the fixed UDA model and compare it against the interpolated label. This comparison determines whether the mixed sample is accurately or inaccurately predicted.
>
> > **Q3**:  L244: Any more justification on why $\lambda=0.55$? How's the performance variation depending on the choice of $\lambda$?
>
> **A3**: The mix ratio $\lambda$ in mixup controls the level of ambiguity of mixed samples. A common heuristic is that a value close to 0.5 for $\lambda$ promotes more ambiguous in-between samples, potentially possessing stronger discriminatory capabilities. In contrast, a value approaching 1.0 generates simpler samples with lower discriminatory potential. This principle is supported by the results from Figure 3, where performance at $\lambda=0.9$ is notably suboptimal. Hence, we set $\lambda=0.55$ for all our experiments to ensure stable validation performance.
>
> > **Q4**:   without looking at the codes and theoretical justifications, the reproducibility cannot be fully verified
>
> **A4**: Our MixVal approach is straightforward, requiring no extra model re-training or extensive hyperparameter tuning. We've included detailed PyTorch-style pseudocode in Appendix A, covering every algorithm detail and step. Furthermore, we'll release our full code, including model training and evaluation, to ensure robust reproducibility of our results.

---

> > ### Comment · Reviewer_zt6b · 2023-08-13
> >
> > I appreciate the authors taking the time to answer all questions during the rebuttal. My questions and concerns are adequately addressed. I keep my score as-is.

---

> > > ### Author Response · Authors · 2023-08-13
> > > **Thanks for your support**
> > >
> > > Thank you for your exceptionally prompt feedback and unwavering support of our paper. We are glad to observe that our responses have effectively resolved your concerns. We are committed to integrating the suggestions from all reviews into our revised manuscript and ensuring the reproducibility of our work through the release of our code. Your invaluable input has undeniably enhanced the quality of our paper, and we sincerely thank you for your dedication and time.

---

### Official Review · Reviewer_PTRC · 2023-07-05

**Soundness:** 3 good
**Presentation:** 3 good
**Contribution:** 2 fair
**Rating:** 5
**Confidence:** 4

**Summary:**

Validating hyperparameters in UDA is challenging due to the unlabeled target data. For it, the paper proposes MixVal, a novel target-only approach that utilizes mixup to synthesize target samples for validation. MixVal combines inductive biases from prior approaches through intra-cluster and inter-cluster mixup, achieving state-of-the-art performance across 11 UDA methods and 4 adaptation settings.

**Strengths:**

1. Validation is crucial in UDA to utilize it in real-world problems, and this paper proposes an effective method to tackle it.
2. Experimentally, it is clearly robust and effective to model selection across multiple benchmarks.

**Weaknesses:**

1. I can't find methodological novelty. I think that the proposed method is a combination of Entropy, SND, and Mixup.
2. There are several papers utilizing Mixup in UDA. It requires not simply mentioning those papers, but rather providing a detailed summary and the differences from them. (e.g., CoWA-JMDS [1])
3. Since Mixup is used for intra-cluster and inter-cluster augmentation, it is possible to utilize other famous Mixup variants like Manifold Mixup [2] or CutMix [3]. However, there are no relevant experiments conducted in this regard.

[1] Lee, Jonghyun, et al. "Confidence score for source-free unsupervised domain adaptation." International Conference on Machine Learning. PMLR, 2022.

[2] Verma, Vikas, et al. "Manifold mixup: Better representations by interpolating hidden states." International conference on machine learning. PMLR, 2019.

[3] Yun, Sangdoo, et al. "Cutmix: Regularization strategy to train strong classifiers with localizable features." Proceedings of the IEEE/CVF international conference on computer vision. 2019.

**Questions:**

Please provide an in-depth response regarding the weaknesses mentioned above.

**Limitations:**

This paper addresses the important research topic, model/hyperparameter validation in UDA. I think that this work can be meaningful to the UDA community. However, it is unclear whether the proposed method is the most optimal approach for validation and even as a work of using Mixup.

---

> ### Author Rebuttal · Authors · 2023-08-10
>
> Thanks a lot for the constructive comments. We have addressed all of your concerns in a detailed manner as outlined below.
>
> > **Q1**:  I can't find methodological novelty. I think that the proposed method is a combination of Entropy, SND, and Mixup.
>
> **A1**:  We **respectfully disagree with this assertion**. Our MixVal methodology stands apart from both Entropy and SND. Unlike Entropy and SND, which directly measure overall raw predictions across all target samples, MixVal takes **a distinct approach by using mixed samples to actively probe the learned target structure**. Furthermore, MixVal introduces **a novel analysis of high-level inductive bias** that differentiates it from existing works in UDA model selection.
>
> Regarding the use of mixup, we acknowledge that it's a fundamental data augmentation technique widely applied in various tasks, as detailed in Section 3.2. **The inclusion of mixup does not necessarily undermine novelty, especially considering its extensive influence—cited over 7200 times.** We emphasize that **the standard mixup is typically applied during the training stage with a mix ratio close to 1.0** ($\lambda \in \text{Beta}(\alpha, \alpha), \alpha \in [0.1, 0.4]$) for regularization effect, and tends to suffer from **performance degradation when the mix ratio is near 0.5 due to manifold intrusion [A, B]**.
> In contrast, our **MixVal harnesses mixup during the inference stage to explore the learned target structure**. This is executed with a fixed mix ratio close to 0.5. **These differences significantly set MixVal apart from other techniques including Entropy, SND, and mixup.**
>
> **From the perspective of experimental performance**, we provide a direct comparison **between Entropy+SND and our MixVal in Table 1 of the attached PDF in the global rebuttal**. Our findings reveal that **the combination of Entropy and SND fails to** demonstrate superior performance over either individual method, while consistently, **MixVal significantly outperforms Entropy+SND** across various tasks.
>
> > **Q2**: There are several papers utilizing Mixup in UDA. It requires not simply mentioning those papers, but rather providing a detailed summary and the differences from them. (e.g., CoWA-JMDS [1])
>
> **A2**:  We appreciate your reference to CoWA-JMDS [1]; we will certainly integrate it into our discussion on related research. It's important to emphasize that **our submission markedly differs from this specific paper**. While the mentioned work contributes to enhancing model performance within UDA, our focus centers on the often-overlooked yet significant model selection challenge in UDA. We've already included references to numerous UDA and semi-supervised learning papers that employ Mixup in L132. Additionally, we have **conducted comprehensive experiments employing MixVal for model selection with the UDA method DMRL [32]**, providing validation results in Table 8 and thorough analysis in L286-L289. This experiment demonstrates that our MixVal approach remains effective for UDA methods utilizing mixup for model training.
>
> > **Q3**: Since Mixup is used for intra-cluster and inter-cluster augmentation, it is possible to utilize other famous Mixup variants like Manifold Mixup [2] or CutMix [3]. However, there are no relevant experiments conducted in this regard.
>
> **A3**:  We appreciate your insights. To clarify, **we've already conducted experiments wherein we substituted mixup with Manifold Mixup and CutMix**. We provide a comparison of the **results in Figure 2 (a) and detailed analysis in L274-L278**. Notably, our findings indicate that image-level mixup outperforms other consistency regularizations including Manifold Mixup and CutMix.
>
> [A] MixUp as Locally Linear Out-Of-Manifold Regularization, AAAI 2019
>
> [B] On Mixup Training: Improved Calibration and Predictive Uncertainty for Deep Neural Networks, NeurIPS 2019

---

> > ### Comment · Reviewer_PTRC · 2023-08-20
> >
> > Thanks the authors for feedback. In terms of novelty and contribution, the proposed concept still appears to be a synthesis of Entropy, SND, and Mixup. While I am not averse to favorable assessments, my evaluation will remain same.

---

> > > ### Author Response · Authors · 2023-08-20
> > > **Addressing your remaining concern on novelty**
> > >
> > > Thanks for the feedback. Regarding your remaining concern "In terms of novelty and contribution, the proposed concept still appears to be a synthesis of Entropy, SND, and Mixup," we aim to address this comprehensively.
> > >
> > > ---
> > > ---
> > >
> > > ### **Our contributions**
> > >
> > > - Our paper addresses the **fundamental problem** of unsupervised model selection in domain adaptation, which remains **underexplored and open**.
> > >
> > > - We are **the first** to tackle this challenge by **directly investigating the structure** learned for the unlabeled target domain. Specifically, we introduce a **novel target-only** model selection method named **MixVal**, which employs mixup for **both inter-cluster and intra-cluster probing**.
> > >
> > > - In comparison to existing studies like SND [27] and Entropy [44], we offer **much more extensive experimental results**. These results **demonstrate** that our method, **MixVal**, stands out as **the only stable approach** consistently achieving **state-of-the-art** performance across a variety of tasks.
> > >
> > > We appreciate the **reviewer's acknowledgment of the two contributions (1 & 3)** made by our paper and note that there remains a concern solely regarding the second contribution, regarding our method MixVal.
> > >
> > > ___
> > > ___
> > >
> > > ### **Our novelties**
> > >
> > > #### **Regarding mixup,**
> > > it's important to emphasize that mixup just serves as a technique for generating probing samples within our model selection method, MixVal. However, **the novelty of MixVal extends far beyond the utilization of mixup**. Still, the **clear differences** between **how we use mixup and the training-stage mixup [30]** are explained as follows:
> > >
> > > - **Different purpose and stage**: In **mixup [30]** and its subsequent applications [31, 32, 33], mixup functions as **a regularization technique** applied during the **training stage** to train a model. Conversely, in **our paper**, mixup operates as **a probing sample synthesis technique** and is integrated into MixVal during the **inference stage** to evaluate a model.
> > >
> > > - **Different mixup strategy**: Many existing applications of mixup **[30, 31, 32, 33] primarily use inter-cluster mixup**, focusing on mixing samples of different classes. In contrast, driven by our novel motivation of probing, **we** are the first to **explicitly** consider and **differentiate** between **inter-cluster** mixup and **intra-cluster** mixup for model selection.
> > >
> > > - **Different mix ratio $\lambda$**: For all existing training-stage mixup applications **[30, 31, 32, 33]**, the mixup operation is **only effective** with a mix ratio **$\lambda$ near 1**, while **ineffective** with a mix ratio **$\lambda$ near 0.5** due to the **manifold intrusion problem [A, B] in our rebuttal**. In contrast, our **MixVal** method demonstrates, through Figure 3 experiments, that a mix ratio **$\lambda$ near 1 is ineffective**, whereas a mix ratio **$\lambda$ near 0.5** yields **better** results.
> > >
> > > ---
> > > #### **Regarding SND [27] and Entropy [44],**
> > > we demonstrate the novelty of our MixVal over these competing model selection methods as follows.
> > >
> > > - **Novel analysis**: As shown in **Table 1**, we introduce a fresh and comprehensive analysis of the **inductive bias used in** existing target-only validation methods **[26, 27, 44, 45]**. This brings forth a **new insightful understanding** of these methods within the domain adaptation **community**.
> > >
> > > - **Novel methodology**: We are **the first** to solve the model selection problem in UDA from **a new probing perspective**. Our MixVal method employs two types of mixed samples to probe the trained model, inherently considering **two** advantageous inductive biases for the **first time**. In contrast, **based on our novel analysis**, both **SND and Entropy** utilize **raw target samples** to evaluate a **singular** inductive bias.
> > >
> > > - **Novel performance**: Compared with Entropy and SND, we significantly **broaden the empirical evaluation scope** to include open-partial-set DA and source-free DA for **the first time**. In addition, Our **MixVal consistently surpasses both in model selection stability and performance**. As suggested by the reviewer, we contrast MixVal with the combined method **"Entropy+SND" in Table 1 of our rebuttal PDF.** The results indicate that the combination of Entropy and SND falls short of improving over either alone. In contrast, **MixVal consistently outperforms Entropy, SND, and "Entropy+SND"** across diverse tasks. This **strongly addresses the concern that "the proposed method is a combination of Entropy, SND".**
> > >
> > > We summarize the MixVal vs. "Entropy+SND" comparison as follows, with complete results available in Table 1 of our rebuttal PDF.
> > >
> > > | Method | ATDOC [25] | BNM [22] | PADA [15] | SAFN [21] |
> > > | :-----| ----: | :----: | ----: | :----: |
> > > | Entropy + SND | 62.16 | 56.26 | 61.27 |  70.52 |
> > > | MixVal (ours) | **68.26** | **66.30** | **67.57** |  **71.41** |
> > >
> > > ---
> > >
> > > **Based on the compelling evidence presented above, we firmly support the novelty and contribution of our paper.**

---

### Official Review · Reviewer_4S59 · 2023-07-07

**Soundness:** 3 good
**Presentation:** 2 fair
**Contribution:** 2 fair
**Rating:** 5
**Confidence:** 3

**Summary:**

In this paper propose a novel target-only method is proposed to employ mixup to synthesize in-between target samples for validation. MixVal leverages mixed target samples to directly probe the learned target structure, benefiting from an combination of inductive biases considered in prior approaches. MixVal performs intra-cluster and inter-cluster mixup to explicitly capture both inductive biases of neighborhood consistency and low density separation.

**Strengths:**

1. This work propose a novel solution named MixVal that eliminates the need for source data access and avoids the cumbersome model re-training. Through inter-cluster and intra-cluster mixup, MixVal combines two essential inductive biases utilized in prior validation methods.
2. The experiment results achieve SOTA.

**Weaknesses:**

1. The presentation need to be improved.
2. Using Mixup in domain adaptation is a common practice.

**Questions:**

DANCE (NeurIPS 2020) is not the SOTA of Open-partial-set UDA. How about the performance of some later work as listed below?
[1] Guangrui Li et al. Domain Consensus Clustering for Universal Domain Adaptation. CVPR2021
[2] Kuniaki Saito et al. OVANet: One-vs-All Network for Universal Domain Adaptation. ICCV2021
[3] Liang Chen et al. Evidential Neighborhood Contrastive Learning for Universal Domain Adaptation. AAAI2022
[4] Liang Chen at al. Geometric Anchor Correspondence Mining with Uncertainty Modeling for Universal Domain Adaptation. CVPR2022

**Limitations:**

The authors addressed the limitations.

---

> ### Author Rebuttal · Authors · 2023-08-10
>
> We appreciate the positive feedback on our paper, particularly regarding its soundness and state-of-the-art experimental results. We have addressed your concerns as follows:
>
>
> > **Q1**:  The presentation need to be improved.
>
> **A1**: Thanks for the constructive suggestion. We will revise our paper according to comments from all reviewers to improve our presentation.
>
> > **Q2**:  Using Mixup in domain adaptation is a common practice.
>
> **A2**: We acknowledge the widespread use of Mixup in various domain adaptation methods. **However, it's important to clarify that our implementation of mixup differs significantly from the conventional practice employed in most mixup-related works.** Numerous domain adaptation approaches [32, 49, 50], as well as semi-supervised learning methods [31, 33], typically apply mixup during the training stage with a mix ratio close to 1.0 ($\lambda \in \text{Beta}(\alpha, \alpha), \alpha \in [0.1, 0.4]$) for regularization effect, and tend to suffer from performance degradation when the mix ratio is near 0.5 due to manifold intrusion [A, B].
> In contrast, our MixVal harnesses mixup during the inference stage to explore the learned target structure. This is executed with a fixed mix ratio close to 0.5.
>
> We've highlighted some representative methods employing mixup for domain adaptation [32, 49, 50] and semi-supervised learning [31, 33] in L32. Furthermore, we've conducted validation experiments, specifically with DMRL [32], presenting results in Table 8 and corresponding analysis in L286-L289. The outcomes from the DMRL experiments demonstrate that MixVal offers resilience against attacks that involve mixup during domain adaptation model training.
>
> We'll also clarify the distinctions between these methods and our approach in our paper.
>
> > **Q3**:  DANCE (NeurIPS 2020) is not the SOTA of Open-partial-set UDA. How about the performance of some later work as listed below?  [1] Guangrui Li et al. Domain Consensus Clustering for Universal Domain Adaptation. CVPR2021 [2] Kuniaki Saito et al. OVANet: One-vs-All Network for Universal Domain Adaptation. ICCV2021 [3] Liang Chen et al. Evidential Neighborhood Contrastive Learning for Universal Domain Adaptation. AAAI2022 [4] Liang Chen at al. Geometric Anchor Correspondence Mining with Uncertainty Modeling for Universal Domain Adaptation. CVPR2022
>
> **A3**:  We greatly appreciate your valuable suggestion. We'll include all of these relevant works in the related section of the open-partial-set UDA. It's worth noting that our submission is the first to tackle the model selection problem in the context of open-partial-set UDA, showcasing the validation method's generalization ability.
>
> Our choice of DANCE as the method for open-partial-set UDA serves specific purposes. Notably, DANCE holds a notable status as a standard universal domain method and has also been adopted by SND [27] for closed-set validation experiments. However, we acknowledge that DANCE doesn't represent the state-of-the-art (SOTA) in open-partial-set UDA.
>
> **To further address the concern on this, we've included the results of OVANet, as recommended by the reviewer, and have presented results in Table 2 of the attached PDF in the global rebuttal.** Specifically, we conduct the hyperparameter validation for the entropy objective's loss coefficient, spanning a range of values such as $\\{0.001, 0.003, 0.01, 0.03, 0.1, 0.3, 1.0, 3.0, 10.0\\}$. OVANet sets the default value as 0.1 by conducting supervised validation on one UDA task. From the results, we observed that MixVal consistently outperforms other target-only validation methods in terms of validation performance. This experiment further highlights the robust generalization ability of MixVal.
>
>
> [A] MixUp as Locally Linear Out-Of-Manifold Regularization, AAAI 2019
>
> [B] On Mixup Training: Improved Calibration and Predictive Uncertainty for Deep Neural Networks, NeurIPS 2019

---

> > ### Comment · Reviewer_4S59 · 2023-08-18
> >
> > I would like to thank the authors for their answers to my questions. Most of my questions have been solved. I choose to keep my rating score.

---

> > > ### Author Response · Authors · 2023-08-18
> > > **Thanks for your support**
> > >
> > > We extend our sincere gratitude to the reviewer for providing prompt and valuable feedback during the discussion phase. We are pleased to learn that our rebuttal has effectively addressed your questions. All of your constructive suggestions will be thoughtfully incorporated into the revised version of our manuscript. Your insightful review has undeniably contributed to improving the quality of our paper, and we genuinely appreciate your dedication and time.

---

### Official Review · Reviewer_G1xN · 2023-07-21

**Soundness:** 2 fair
**Presentation:** 3 good
**Contribution:** 2 fair
**Rating:** 5
**Confidence:** 4

**Summary:**

This paper introduces a simple validation method for unsupervised domain adaption. This method, called MixVal, proposes a new evaluation (ICE) based on MIXUP to select the most appropriate model candidates, which is obtained by training with different hyperparameters, such as loss coefficient/temperature/margin factor. Compared with other target-only validation methods, the paper considered three different inductive biases: neighborhood consistency/low-density separation/no prior label distribution.

**Strengths:**

This paper is well-written and easy to follow. Specifically, Table 1 and Figure 1 clearly show the motivation and pipeline, respectively.

Experiments over several datasets and settings are impressive which sufficiently show the generalization of the proposed method.

**Weaknesses:**

**1. Lack novelty.** Behind the mentioned contributions, there is only one proposed technical point, the proposed evaluation ICE with MIXUP, which just takes less than half a page. It could not be enough as a NeurIPS paper, especially lacking sufficient theoretical support.

**2. Lack theoretical support.** The method assumes that the model with a higher ICE score can obtain a higher real target accuracy. This paper needs such theoretical support that there is a positive correlation between the ICE score and the discriminability of the candidate models.

This assumption could not be valid in real-world systems. Ideally, the best ICE score is 100%. In this case, the “model” could be a linear system, which is certainly impossible in the applications. This might demonstrate there are some theoretical gaps between the score and the goal of UDA.

The authors also recognized this limitation and tried to give some empirical observations in Figure 2(b). However, without the second technical point, this limitation seems very serious.

Figure 2(b) looks a little bit tricky, where the first figure is shown with accuracy, and the following subfigures are in the ranked ascending order. In fact, all subfigures should be shown with the same indicator, ranking all model candidates in ascending order. In this case, the empirical conclusion about the high correlation is not convincing.

**3. Confusion about formulation.** The formulation of the mixed pseudo label is different from its pseudocode.

**Questions:**

What are the relationships between RankInter and RankIntra? Is that necessary for us to separate such two different types during inference?

In Figure 3, AccAvg has a similar trend as RankIntra. RankInter and RankIntra seem not to compensate. In other words, handling the inductive bias neighborhood consistency and low-density separation at the same time does not make sure higher performance.

**Limitations:**

Please see the weakness.

---

> ### Author Rebuttal · Authors · 2023-08-10
>
> Thanks for the constructive comments. We've provided responses to each of your remaining questions below.
>
> > **Q1**:  Lack novelty. Behind the mentioned contributions, there is only one proposed technical point, the proposed evaluation ICE with MIXUP, which just takes less than half a page. It could not be enough as a NeurIPS paper, especially lacking sufficient theoretical support.
>
> **A1**:   We appreciate the reviewer's recognition of **MixVal's simplicity and the novelty introduced by our proposed ICE evaluation with Mixup**. However, we **respectfully differ** on the assertion that our submission **lacks novelty due to a single proposed technical point**. Firstly, we'd like to clarify that the **ICE with Mixup encompasses two novel technical aspects** introduced by our submission: **the validation of models via synthesized sample probing and the novel application of mixup during the inference stage** to generate in-between discriminative samples. The simplicity of ICE might overshadow these nontrivial innovations.
>
> Beyond the technical facets, our innovation extends to the **novel analysis of high-level inductive bias** inherent in existing target-only validation methods, succinctly summarized in Table 1. Furthermore, **extensive empirical comparisons spanning diverse domain adaptation scenarios** are presented in Tables 2-7.
>
> As for the content coverage, our introduction of MixVal actually spans **more than one page, as shown in Section 3.2**. As for the assertion that a submission with less than half a page of technical content may not suffice for NeurIPS, we respectfully differ in opinion.
>
>  > **Q2**:   Lack theoretical support. ... This paper needs such theoretical support that there is a positive correlation between the ICE score and the discriminability of the candidate models. This assumption could not be valid in real-world systems. ... This might demonstrate there are some theoretical gaps between the score and the goal of UDA.
>
> **A2**:   Although formal theoretical analysis is absent, our **extensive experiments and analysis strongly validate MixVal**. Across various UDA scenarios outlined in Tables 2-7 (including VisDA, DomainNet, source-free UDA, and open-set shifts), MixVal consistently outperforms existing validation methods. This compelling empirical evidence contests the notion that "this assumption could not be valid in real-world systems." Our results affirm MixVal's effectiveness in model selection based on ICE scores, **dispelling any unknown but conjectured theoretical gaps.**
>
>  > **Q3**: ... some empirical observations in Figure 2(b). ... Figure 2(b) looks a little bit tricky, In this case, the empirical conclusion about the high correlation is not convincing.
>
> **A3**: We'd like to clarify that **Figure 2(b) is informative with comprehensive data**. While we can infer the rank from specific accuracies in the first part of Figure 2(b), reversing the process is not feasible. To enhance clarity, we've also included a ranking visualization in **Figure 1 of the PDF. Our observation remains consistent: MixVal exhibits a stronger correlation than the state-of-the-art method SND.**
>
>  > **Q4**: Confusion about formulation. The formulation of the mixed pseudo label is different from its pseudocode.
>
> **A4**: No inconsistency between method and pseudocode. **Mixup formulation is general, while pseudocode offers detailed steps.**
>
>  > **Q5**: What are the relationships between RankInter and RankIntra? Is that necessary for us to separate such two different types during inference? In Figure 3, AccAvg has a similar trend as RankIntra. RankInter and RankIntra seem not to compensate. In other words, handling the inductive bias neighborhood consistency and low-density separation at the same time does not make sure higher performance.
>
> **A5**: To clarify, "RankInter" signifies MixVal with inter-cluster mixup only, and "RankIntra" corresponds to MixVal with intra-cluster mixup only. In our MixVal implementation, we **achieve both types of probing seamlessly in a single-time inference**, as outlined in our "ice_score" function detailed in Appendix A.
>
> Regarding Figure 3, it's important to note that AccAvg is not employed in our MixVal methodology. This is because the direct combination of ICE scores from the two probing strategies is unstable, given that intra-cluster probing tends to yield higher ICE scores than inter-cluster probing. The Figure illustrates that AccAvg performs less effectively than RankAvg due to the instability associated with accuracy averaging. In our submission, **we highlighted in L298 that MixVal's combination of both probing strategies bolsters the stability of model selection, substantiated by our experimental outcomes detailed in Tables 2-5. In support of this point, we've provided additional results in Table 1 of the attached PDF in the global rebuttal.**
>
> Two conclusions can be drawn from these results: (i) The two-dimensional probing strategy enhances the stability of validation compared to individual probing types. (ii) MixVal distinguishes itself from and significantly surpasses the combination of SND and Entropy.

---

> > ### Comment · Reviewer_G1xN · 2023-08-16
> > **Confusion about formulation**
> >
> > Thank the authors for the detailed response. However, I am still confused about the formulation of the mixed pseudo-label. In the main paper, the formulation behind Line 187 shows that $y_{mix}$ is a mixing value of pseudo labels of different target samples, where $\lambda$ is the mixing weight. However, in Algorithm 1, ''mix_labels'' is "pl_a" if "$\lambda$>0.5" and "pl_b" otherwise. Could the author provide some explanations about this?

---

> > > ### Author Response · Authors · 2023-08-16
> > > **Addressing your remaining confusion**
> > >
> > > **Thank the reviewer for providing a detailed description of the confusion about the formulation of the mixup (highlighted as weakness#3 in the review and addressed in Q4&A4 of our rebuttal).** We aim to address this confusion with the following elucidation:
> > >
> > > To ensure clarity in our explanation, we reiterate the pertinent formulations and explanations outlined in our submission. For reference, we reproduce the formulation from Line 187 below. This formulation is commonly employed in mixup-relevant papers to illustrate the mixup operation [30].
> > >
> > > $x_{mix} = \lambda * x_t^i + (1 - \lambda) * x_t^j $
> > >
> > > $y_{mix} = \lambda * \widehat y_t^i + (1 - \lambda) * \widehat y_t^j$
> > >
> > > As explained in Lines 188-189, $\lambda$ is a scalar used for interpolation, $x_t^i$ and $x_t^j$ denote two different target image vectors and $\hat{y}_t^i$ and $\hat{y}_t^j$ denote the corresponding one-hot pseudo label encodings for both images  (as specified in Lines 142-143), $\hat{y} \in \mathcal{R}^{K}$, $K$ is the number of categories in the source domain.
> > >
> > > **The confusion raised by the reviewer pertains to $y_{mix}$.** We agree that "$y_{mix}$ is a mixing value of pseudo labels of different target samples, where $\lambda$ is the mixing weight." Assuming the target sample $i$ and sample $j$ belong to distinct categories $k_i$ and $k_j$ respectively, $y_{mix}$ is presented as a soft label vector $(..., \lambda,..., 1-\lambda, ...)$, with $\lambda$ assigned to the $k_i$-th position and $1-\lambda$ assigned to the $k_j$-th position.
> > >
> > > In the original mixup framework [30] and its subsequent applications in domain adaptation [32, 49, 50], $y_{mix}$ is directly utilized for model training. In this context, the encoded relationship between different samples contributes to regularizing the model training process.
> > >
> > > **Differently, our MixVal is exclusively implemented during the inference stage.** Here, we employ $y_{mix}$ to evaluate the inference accuracy of a fixed model on mixed samples. As demonstrated in our ICE formulation (behind Line 194), for each mixed sample $i$, we **compute the accuracy between its interpolated label $y_{mix}^i$ and its predicted one-hot label $\hat{y}_{mix}^i$**, inferred by the model.
> > >
> > > **For accuracy calculation, only the hard one-hot versions of the interpolated and predicted labels are needed**, focusing on the class with the highest probability. Consequently, for $y_{mix}^i$, we **determine its hard one-hot label by comparing $\lambda$ at the $k_i$-th position and $1-\lambda$ at the $k_j$-th position**, as the probabilities of other categories are zero. **If $\lambda > 1-\lambda$, i.e., $\lambda > 0.5$**, the one-hot label is attributed to the $k_i$-th category. Conversely, **if $\lambda \leq 1-\lambda$, i.e., $\lambda \leq 0.5$**, the one-hot label corresponds to the $k_j$-th category. **This explains why our pseudocode in Algorithm 1 uses an "if-else statement" that compares $\lambda$ and $0.5$ to determine the hard one-hot label of the interpolated label $y_{mix}$.**
> > >
> > > In summary, we would like to emphasize that the formulation from Line 187 and the ICE formulation behind Line 194 are **completely consistent** with the " 'mix_labels' is 'pl_a' if '0.5' and 'pl_b' otherwise" pseudocodes in Algorithm 1. These pseudocodes precisely reflect our implementation of MixVal.
> > >
> > > We sincerely hope that the above explanation effectively addresses your confusion. We welcome any further discussions and value your input.

---

> > > > ### Comment · Reviewer_G1xN · 2023-08-20
> > > >
> > > > Thank the authors for further explanation! I have no more questions. I will consider increasing my score to "borderline accept". Good luck!

---

> > > > > ### Author Response · Authors · 2023-08-20
> > > > > **Thanks for your support**
> > > > >
> > > > > Thank you for your active involvement and prompt feedback during the discussion phase. It's good to know that our rebuttal has effectively answered your questions. We'll carefully integrate all your valuable suggestions into the updated version of our manuscript. Your constructive review has undeniably enhanced the quality of our paper, and we genuinely appreciate your dedication and time.

---

### Author Rebuttal · Authors · 2023-08-10

We sincerely appreciate the reviewers for their valuable time and feedback. Particularly, we are grateful for the following recognitions:

- Our work investigates a **crucial** [Reviewer PTRC] problem which is **well-defined and well-motivated in practice.** [Reviewer zt6b].
- Our method MixVal is **motivated** [Reviewer G1xN], **novel** [Reviewer 4S59], **simple** [Reviewer zt6b], **effective** [Reviewer PTRC, Reviewer zt6b], and **robust** [Reviewer G1xN, Reviewer PTRC, Reviewer zt6b].
- Our experiments are **impressive** [Reviewer G1xN], **extensive** [Reviewer zt6b] and **state-of-the-art** [Reviewer 4S59] to demonstrate the effectiveness of our method.
- Our work has good soundness [Reviewer 4S59, Reviewer PTRC, Reviewer zt6b],  good presentation [Reviewer G1xN, Reviewer PTRC, Reviewer zt6b], and excellent contribution [Reviewer zt6b].

---

### Decision · Program_Chairs · 2023-09-21

**Decision:**

Accept (poster)

**Comment:**

This paper introduces MixVal, a target-only framework that leverages Mixup to generate mixed samples for validation. While the utilization of mixed samples is not without precedent, its application to the validation of domain-adapted models is a distinctive contribution of this work. MixVal employs intra-cluster and inter-cluster Mixup to capture the inductive biases of neighborhood consistency and low-density separation. Extensive evaluations over various domain adaptation methods and different settings prove the effectiveness and generalization ability of the proposed method.